# Time-varying optimization of COVID-19 vaccine prioritization in the context of limited vaccination capacity

Shasha Han [1,2,10], Jun Cai [3,10], Juan Yang [3,4], Juanjuan Zhang[3], Qianhui Wu[3], Wen Zheng[3], Huilin Shi[3], Marco Ajelli [5,6,11], Xiao-Hua Zhou [1,7,8,11✉] & Hongjie Yu [3,4,9,11✉]

Dynamically adapting the allocation of COVID-19 vaccines to the evolving epidemiological situation could be key to reduce COVID-19 burden. Here we developed a data-driven mechanistic model of SARS-CoV-2 transmission to explore optimal vaccine prioritization strategies in China. We found that a time-varying vaccination program (i.e., allocating vaccines to different target groups as the epidemic evolves) can be highly beneficial as it is capable of simultaneously achieving different objectives (e.g., minimizing the number of deaths and of infections). Our findings suggest that boosting the vaccination capacity up to 2.5 million first doses per day (0.17% rollout speed) or higher could greatly reduce COVID-19 burden, should a new wave start to unfold in China with reproduction number ≤1.5. The highest priority categories are consistent under a broad range of assumptions. Finally, a high vaccination capacity in the early phase of the vaccination campaign is key to achieve large gains of strategic prioritizations.

[1] Beijing International Center for Mathematical Research, Peking University, Beijing, China. [2] Harvard Medical School, Harvard University, Boston, MA, USA. [3] School of Public Health, Fudan University, Key Laboratory of Public Health Safety, Ministry of Education, Shanghai, China. [4] Shanghai Institute of Infectious Disease and Biosecurity, Fudan University, Shanghai, China. [5] Department of Epidemiology and Biostatistics, Indiana University School of Public Health, Bloomington, IN, USA. [6] Laboratory for the Modelling of Biological and Socio-technical Systems, Northeastern University, Boston, MA, USA. [7] Department of Biostatistics, School of Public Health, Peking University, Beijing, China. [8] National Engineering Laboratory of Big Data Analysis and Applied Technology, Peking University, Beijing, China. [9] Department of Infectious Diseases, Huashan Hospital, Fudan University, Shanghai, China. [10] These authors contributed equally: Shasha Han, Jun Cai. [11] These authors jointly supervised this work: Marco Ajelli, Xiao-Hua Zhou, Hongjie Yu. ✉email: azhou@math.pku.edu.cn; yhj@fudan.edu.cn

Vaccination is promising to end the COVID-19 pandemic while allowing restoring social activities[1,2]. However, the anticipated global vaccine capacity in 2021 would not be enough to vaccinate every human being on the planet[3,4]. The situation may be worse if we account for possible failures of vaccine candidates, financing shortfalls, as well as logistical and vaccine administration challenges[4,5], or difficulties in expanding manufacturing capacity[6]. Moreover, because of unavoidable inequalities among countries, more than half of the world's population would probably remain unvaccinated until 2023[7].

A two-dose vaccination campaign has been started in China with vaccines administrated first to essential workers (3.28% of the full populations; Supplementary Table 2), then to the general populations of aged 18–59 years, and later extended to all 18+ individuals[8]. As of May 8 2021, less than 8% of general population has been vaccinated with two doses (317.5 million doses[9]). More than 90% of the Chinese population remains to be vaccinated, calling for possible strategic prioritizations. Although optimal prioritization strategies are estimated to provide a larger reduction of COVID-19 burden compared to a random mass vaccination[10,11], they remain elusive and intrinsically connected to the target of the program (e.g., averting deaths vs. reducing the strain on the healthcare system) designed offline.

In this context, it is of paramount importance for governments to set up effective vaccination campaigns as COVID-19 cases can grow at a far higher pace than immunity accumulates in the population. Therefore, defining time-varying vaccination strategies may be highly beneficial[11,12]. Here we propose a data-driven vaccination model coupled with SARS-CoV-2 transmission to optimize the prioritized allocation of vaccines to averting the largest possible number of infections, symptomatic cases, hospitalizations, ICU admissions, and deaths.

## Results

We consider five risk metrics (number of infections, cases, hospital admissions, ICU admissions, and deaths) and minimize their total incidences. Note that essential workers, corresponding to 3.28% of the total population (Supplementary Table 2), are vaccinated first. We consider 2.0 million first doses of a 2-dose vaccine administrated per day (0.14% rollout speed), according to the number of daily doses administered in China from Mar 1, 2021 to May 8, 2021[9]. Daily allocation decisions are coupled with the transmission dynamics of an epidemic spreading, under the hypothesis that non-pharmacological interventions (NPIs) are capable to keep SARS-CoV-2 reproduction number ($R$) at 1.5 and considering a 2-dose all-or-nothing vaccine where the first dose does not confer protection. The optimization is carried out over a time window of 400 days, roughly corresponding to the duration of a simulated epidemic with no vaccination.

When the vaccination program aims to minimize the number of SARS-CoV-2 infections, we estimate that the optimal strategy prioritizes individuals aged 15–39 year until 46.6% coverage is reached; then, vaccines are administered to individuals aged 40–64 years until 25.7% coverage is reached (Fig. 1 and Supplementary Fig. 15). Different age-prioritizations are identified if the goal is to reduce SARS-CoV-2 severe outcomes. For example, to minimize the number of deaths, individuals aged 65 years and older are identified as the first priority until 100% coverage is reached, followed by aged 40–64 until 97.2% coverage (Fig. 1 and Supplementary Fig. 15). To minimize the number of ICU admissions, first priority is given to individuals aged 65+ years and nearly all of them need to be vaccinated before moving to other age groups (Fig. 1 and Supplementary Fig. 15).

Optimal prioritization strategies, although performing best to achieve their specific goals, are capable of dramatically reducing

COVID-19 burden, preventing 634.3–642.2 million infections (89.7–90.8% reduction), 171.3–172.9 million symptomatic cases (90.3–91.1% reduction), 56.7–57.3 million hospital admissions (90.2–91.2% reduction), 3.6–3.9 million ICU admissions (91.6–93.2% reduction), and 5.9–6.3 million deaths (87.9–93.0% reduction) (Fig. 2a–e). We compare the five optimal prioritization strategies with a uniform strategy (random mass vaccination) where vaccines are allocated proportionally to the size of the unvaccinated susceptible population in each age group. We estimate that the optimal prioritization strategies perform dramatically better than the uniform strategy with respect to any risk metric (more than 87% reduction vs. less than 67%; Fig. 2f).

We conduct a univariate analysis to explore the impact of key parameters on the definition of the priority groups and coverages as well as the benefits of the optimal prioritization strategies. We estimate that, for each optimal strategy, the two age categories with the highest priority are broadly consistent under different hypotheses on vaccination capacity, vaccine efficacy, differential vaccine efficacy in preventing infection and disease, vaccine hesitancy, timing of vaccination campaign relative to epidemic onset, SARS-CoV-2 transmissibility, lower infectiousness of asymptomatic individuals relative to symptomatic ones, and uncertainties in the observed cases (reporting rates and lags). Not only are the identified priority orders identical (See Fig. 1, Supplementary Figs. 3, 6, 13 and 14), but also the associated coverages show only little variations in most cases (Supplementary Fig. 7). However, if the vaccination program aims at minimizing the number of infections, the identified priority orders change when consider age-mixing patterns estimated during the pandemic (as compared to the pre-pandemic mixing patterns used in the baseline analysis; the obtained results are reported in Supplementary Fig. 17b). Moreover, if the vaccine efficacy is identical across all age groups or if the vaccine is administered regardless of a previous history of infection, a high priority is given to adults aged over 65 years also when the vaccination campaign aims at minimizing the number of infections (e.g., Supplementary Figs. 17a and 19a). Finally, should the vaccine rollout end before the onset of an epidemic, the prioritization order does not affect the final outcome, which entirely depends on the vaccine coverage (Supplementary Fig. 20).

Considering $R = 1.5$, we estimate that the advantage of optimal prioritization strategies over a uniform mass vaccination increases with 2.0 million first doses administrated per day (0.14% rollout speed) is reached and decreases as the vaccination capacity further increases. When the capacity is sufficiently high (3.5 million first doses administered per day; 0.24% rollout speed), also a uniform mass vaccination would be sufficient to avert nearly all deaths as compared with an epidemic controlled with NPIs only ($R = 1.5$). The advantage of the optimal strategies with respect to a uniform vaccination remain unaltered for variations in all other parameters regulating the vaccination, namely vaccine efficacy, differential vaccine efficacy in preventing infection vs. disease, vaccine hesitancy, uncertainty on the observed cases (including reporting lags), vaccination capacity regardless of whether an individual was already infected by SARS-CoV-2. On the other hand, the advantage of prioritized strategies decreases the larger the gap between the vaccination campaign starting and the epidemic onset (Fig. 3 and Supplementary Figs. 8–11 and 13).

Considering alternative values of R (and thus different levels of NPIs) greatly affects the effectiveness of these strategies, and thus the relative advantages of optimal strategies to the mass vaccination. The benefits are negligible when $R = 1.25$, as all strategies can reduce the epidemic burden by almost 100%. For R larger than 1.5, the benefits may depend on the program targets, e.g., only those aiming at minimizing deaths and ICUs performing

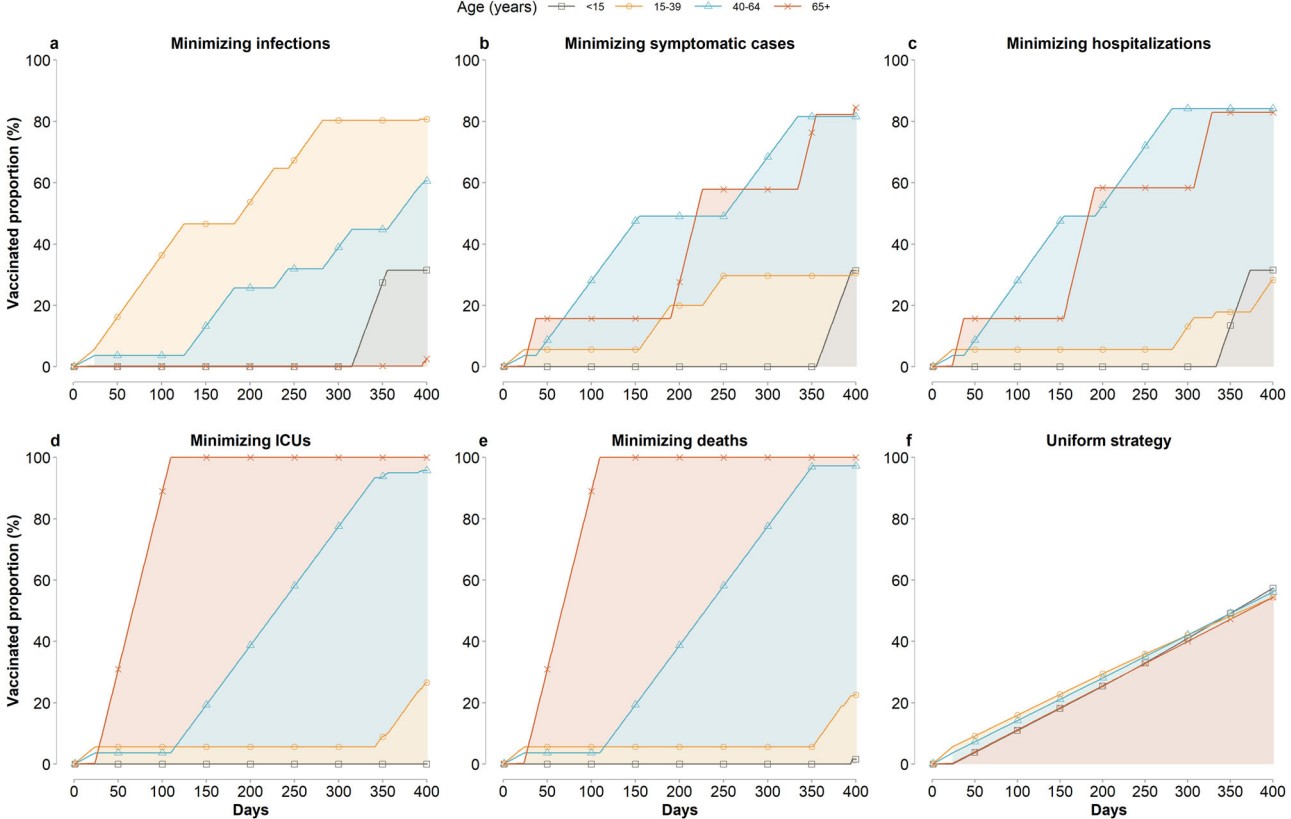

**Fig. 1 Estimated vaccine coverages over time for different prioritization strategies (0.14% rollout speed, _R_ = 1.5). a–e** Five optimal prioritization strategies minimizing the total incidence of infections, symptomatic cases, hospitalizations, ICU admissions, and deaths, respectively. **f** Uniform vaccination strategy. Shaded area refers to vaccination capacity to the general populations. Lines refers to the vaccinated proportions including essential workers. Keys and labels apply to all panels.

markedly better than the random mass vaccination in averting deaths or ICUs (Fig. 3i and Supplementary Fig. 11i).

We also test an alternative vaccine model, replacing the baseline all-or-nothing mechanism (i.e., the vaccine either provides full protection or no protection to individuals, according to the vaccine efficacy): the "leaky vaccine" model, where all vaccinated individuals are exposed to a lower risk of infection corresponding to the vaccine efficacy[13] (see Methods). The "leaky vaccine" model identifies optimal prioritization strategies and effectiveness highly similar to those for the all-or-nothing model as well as quantitatively similar reductions of all risk incidences (Supplementary Figs. 7 and 12).

Furthermore, we investigate the vaccination capacity needed for a prioritized vaccination campaign to keep the total number of deaths under 10,000. We estimate that, for _R_ = 1.5 and an "all-or-nothing" vaccine, the minimal daily vaccination capacity corresponds to 3.5 first doses (0.24% of the population).

## Discussion
Our results show that time-varying COVID-19 vaccination strategies can greatly reduce COVID-19 burden. The optimal vaccination strategies with specific program targets (e.g., minimize the number of deaths) are shown to be driven by the targeted risks of population segments as an epidemic unfolds. Benefits of optimal strategies as a comparison to random mass vaccination is substantial when the capacity is modestly low, or the transmission is modestly high. Nonetheless, the random mass vaccination may potentially represent an alternative to targeted vaccination strategies should the capacity be sufficiently high (Supplementary

Fig. 2a), the transmissibility controlled be at a very low level (Supplementary Fig. 2c), or the vaccine be administered to individuals aged 20–59 years only (Supplementary Fig. 5c)—i.e., the age groups for which the administrated vaccine in the early phase has proved to be safe and efficacious.

Given the uncertainty still surrounding many of the key parameters regulating the vaccination process, we tested several scenarios on vaccination capacity, vaccine efficacy, differential vaccine efficacy in preventing infection vs. disease (Supplementary Fig. 4), vaccine hesitancy of the population (Supplementary Fig. 5) and different timings of the start of the vaccination campaign relative to the epidemic onset (Supplementary Fig. 13). We also tested two alternative vaccine mechanisms ("all-or-nothing" vs. "leaky" vaccine; Supplementary Fig. 12). We found that all those factors have little effect in determining the optimal prioritization strategies. Although this increases the confidence in our findings, it is important to stress that the identified optimal strategies are sensible to the variation in age-mixing contact patterns and the differences in vaccine efficacy by age. This highlights the need to potentially adapt vaccination choices to the implemented NPIs (which may shape age-mixing patterns[14–16]) and the heterogeneity of vaccine efficacy across age groups.

We examined the connection between the daily vaccination capacity (rollout speed) and transmissibility in determining the optimal strategies. In line with previous work[17], we found that differences between prioritization strategies are negligible when the daily capacity is high enough relative to the reproduction number R (Fig. 3a and i, Supplementary Fig. 2a and c). However, we found that the optimal vaccination strategies consistently prioritize high risk population across varying scenarios including

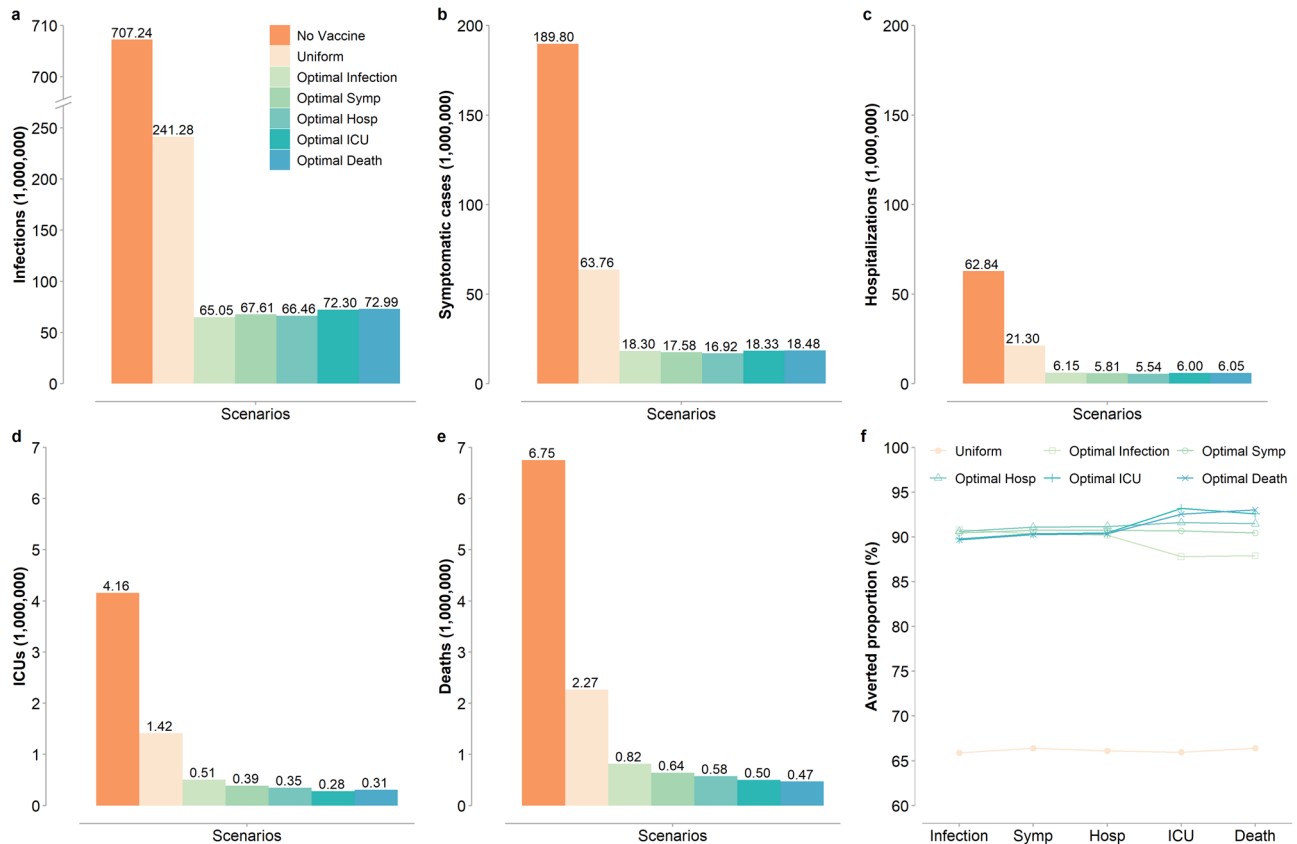

**Fig. 2 Risk incidences for different prioritization strategies and the scenario with no vaccination (0.14% rollout speed, $R = 1.5$). a** Number of infections under the scenario with no vaccination, uniform vaccination strategy, and the five optimal prioritization strategies we identified. **b–e** As (**a**), but for the number of symptomatic cases, hospitalizations, ICUs, and deaths, respectively. **f** Averted proportion of infections, symptomatic cases, hospitalizations, ICUs, and deaths for the six vaccination strategies as compared to the scenario with no vaccination. Keys in **a** apply to (**a–e**). "Optimal Symp" and "Optimal Hosp" represent the two optimal strategies for minimizing symptomatic cases and hospitalizations respectively.

different rollout speeds, SARS-CoV-2 transmissibility, and vaccine mechanisms, although, depending on the scenario (e.g., Fig. 1, Supplementary Figs. 3a, 7 and 12b). This novel finding is likely associated with the methodology we used, which allowed us to consider strategies where the vaccination can target a new age group before the full coverage in the previous group is reached.

Our findings suggest that boosting the daily capacities into 2.5 million first doses (0.17% rollout speed) or higher could greatly reduce COVID-19 burden should a new wave start to unfold in China. Moreover, we estimate that a high vaccination capacity in the early phase of the vaccination campaign is key to achieve large gains of strategic prioritizations. All strategies result in a much larger disease burden when vaccine rollouts gradually increase over time (although the same total amount of doses is administered; Supplementary Fig. 16). Furthermore, we found that vaccinating individuals regardless of their infection history has a relatively small impact on the epidemic burden (Supplementary Fig. 19b), although the identified prioritization strategies differ from that in the baseline (Supplementary Fig. 19a). This finding could be associated with the vaccination capacity of the Chinese health system and the considered value of the reproduction number (i.e., 1.5) and it thus not necessarily be the case in other contexts[17].

The timing of vaccination relative to the epidemic onset plays a crucial role in determining the effectiveness of prioritization strategies. Indeed, should a large enough fraction of the population already be vaccinated before an epidemic starts to unfold, the effectiveness of prioritized strategies is similar to that of a random vaccination (Fig. 3g and Supplementary Fig. 13).

When vaccines have low efficacy (60%) in preventing the infection, the effectiveness of optimal prioritization strategies is stable (Fig. 3e), while the impact on reducing COVID-19 burden increases with the efficacy in preventing the disease (Supplementary Fig. 4a–c). On the other hand, if the vaccine is protective against the disease only, prioritized strategies can still be effective in reducing the total number of patients requiring an ICU and the total number of deaths (Fig. 3f, Supplementary Figs. 4d and 11f).

The identified prioritization strategies and the estimated effectiveness of the vaccination campaign are consistent when accounting for uncertainty in case reporting and delays (Fig. 3l, Supplementary Fig. 14).

It is key to remark that our results are obtained by assuming $R = 1.5$, with NPIs implemented throughout the entire vaccine rollouts. As a consequence, our results neither provide estimates of the herd immunity threshold for COVID-19 nor can be used to estimate the overshoot of the epidemic after the herd immunity threshold is reached.

Our study adds to the literature in several ways. First, although investigating time-varying vaccination program is not novel, to the best of our knowledge, this is the first investigation at the population level and in a data-driven context accounting for estimates of vaccination capacity. Time-varying vaccination allows the campaign to track the eligible population and adapt the targeted vaccination populations to the evolving epidemiological situation, and thus protect individuals who are at the highest risks at each time. Second, previous studies have shown that optimal prioritization strategies for single objectives (e.g., minimize the number of death) may sacrifice secondary objectives (e.g.,

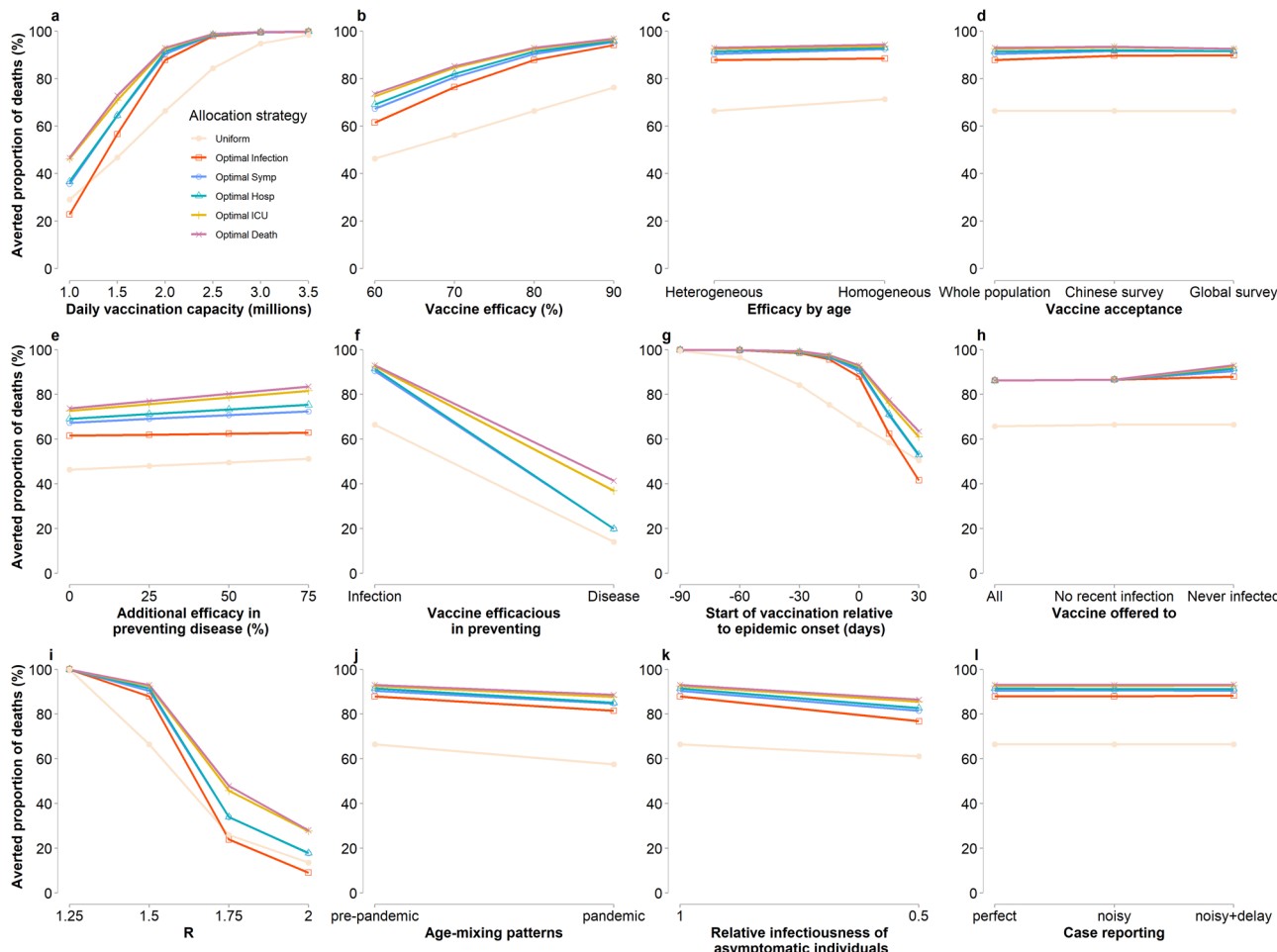

**Fig. 3 Effectiveness of alternative vaccination strategies under alternative assumptions. a** Averted proportion of deaths for different strategies (uniform vaccination and the five optimal strategies) as compared to no vaccination for different values of the vaccination capacity. **b** As (**a**), but for different values of the vaccine efficacy. **c** As (**a**), but assuming vaccine efficacy to be heterogeneous by age ("Heterogeneous", as in the main analysis) or homogeneous ("Homogeneous") at 80% across age groups. **d** As (**a**), but considering possible vaccine hesitancy. In particular, we consider three alternative scenarios: (i) "Whole population", i.e., all individuals are willing to accept the vaccine; (ii) "Chinese survey", i.e., on average 83% of the population is willing to accept the vaccine as estimated in a survey conducted on the Chinese population (age-specific estimates reported in Supplementary Table 2); (iii) "Global survey", i.e., on average 61% of the population is willing to accept the vaccine as estimated in a survey at the global level[41]. **e** As (**a**), but assuming that the vaccine has 60% efficacy in preventing the infection and 0%, 25%, 50%, or 75% efficacy in preventing the disease given the infection. **f** As (**a**), but without the strategy minimizing the incidence of infections and for different vaccines actions, (i) "Infection", i.e., 80% efficacy in preventing infections as in the main analysis, and (ii) "Disease", i.e., 80% efficacy in preventing disease and 0% efficacy in preventing infections. **g** As (**a**), the vaccination campaign starts between 90 days before and 30 days after the epidemic onset. **h** As (**a**), but the vaccine is administered to (i) all individuals regardless of their history of infection ("All"); (ii) only to individuals who were not infected in the previous 6 months ("No recent infection"); and (iii) only to individuals who were never infected ("Never infected"). **i** As (**a**), but for different values of the reproduction number. **j** As (**a**), but using "pre-pandemic" age-mixing patterns (as in the baseline analysis) or "pandemic" mixing patterns. **k** As (**a**), but assuming that the infectiousness of asymptomatic individuals relative to symptomatic individuals is 1 (as in the baseline) or 0.5. **l** As (**a**), but we consider the reporting of cases to be: (i) perfect ("perfect", as in the baseline analysis); (ii) subject to noise ("noisy"); or subject to noise and an 11-day delay ("noisy+delay"). Results for other risk metrics are similar and displayed in Supplementary Figs. 8–11. Keys and labels apply to all panels. "Optimal Symp" and "Optimal Hosp" represent the two optimal strategies for minimizing symptomatic cases and hospitalizations respectively.

minimize the number of infections)[18]. Instead, our results show that, thanks to the time-varying vaccination, the sacrifices are lower than previously estimated. Indeed, the time-varying optimal vaccination can account for changes in age-specific risks (e.g., of infection, hospitalization) over time. This finding supports the relevance of both direct and indirect protection of the population[19] to define prioritized vaccination strategies. Our findings are of particular relevance for China at this early phase of vaccine rollout when vaccination is restricted to people mainly aged 18–59, due to the lack of vaccine safety and efficacy thus far.

To properly interpret our findings, it is important to consider the limitations of the performed analysis. First, the analysis is based on a deterministic model and does not consider the stochasticity of the real-world infection transmission process, although the former can be considered as a good approximation of the latter when the number of infections is large. In the early phase of the epidemic, the stochastic variability may affect the timing of the epidemic peak, resulting in inaccuracy in the estimated effectiveness of vaccination programs. Also, we consider average values of model parameters (e.g., the contact matrix,

age-specific estimates of the susceptibility to infection). Though we recognize lack of estimates of the uncertainty around our point projections, we have conducted extensive sensitivity analysis showing to what extent model parameters affect the obtained results. Second, we leveraged contact data collected before the COVID-19 pandemic to model SARS-CoV-2 transmission through 2021 and accounted for the impact of NPIs on contact numbers through a reduction on R, but the future contact pattern by age remains elusive. Moreover, we used the contact pattern data collected in Shanghai and extrapolated it into the entire China. Although this surely represents a limitation, several independent studies showed little variation in age-mixing patterns across China[15,20,21]—i.e., individuals tend to have more contacts within individuals of similar ages. When this age-assortative pattern is less pronounced, benefits of optimal prioritization relative to the random mass vaccination are smaller (Supplementary Fig. 18a). The benefits would also be smaller if the overall transmissions are higher even with similar age-mixing patterns (would symptomatic cases are twice as infectiousness as asymptomatic cases; Supplementary Fig. 18b). Third, for model tractability, we optimize the allocation of the first dose of the two-dose vaccine only, and then require the second dose to be administrated following the vaccination schedule, although this may not be optimal from the mathematical point of view. Fourth, the value of our identified time-varying optimal prioritization strategies lies in scenarios when vaccination campaign continues as epidemic unfolds. If a campaign is fully completed before epidemic, the optimal allocation strategies with different program targets are similar without priority sequencing (Supplementary Fig. 20). Furthermore, our study solely focuses on pursuing time-varying vaccination strategies that timely adapt to the evolved situation of a predictable epidemic. However, when the evolution of epidemic per se cannot be forecasted in advance, epidemiologically-driven prioritization strategies that adapt with both the time and the realized state as the epidemic unfolds may have larger gains. Finally, although our analysis takes into account practical constraints such as the prioritization of essential workers and the exclusion of adults over 60 years of age (Supplementary Figs. 5c and 6c), other ethical considerations should be taken into account as well.

In sum, our analysis identified optimal COVID-19 vaccination prioritizations as the epidemic unfolds. Our model-based evaluation highlights the benefit of these strategies in simultaneously minimizing different objectives (e.g., number of deaths and infections). Finally, the modeling framework presented here is general enough to be adopted by other countries to identify optimal vaccine prioritization strategies conditional on the country-specific socio-demographic features, evolving epidemiological situation, vaccination capacity, and other factors (ethical, political, and societal) in practical implementation.

## Methods

**Model**. We model SARS-CoV-2 transmission dynamics using an age-structured compartmental model with 17 age groups {0–4,5–9,10–14,15–19,20–24,25–29,30–34,35–39,40–44,45–49,50–54,55–59,60–64,65–69,70–74,75–79,80+; total number of age groups denoted with $J$}. For each age group, the population is divided into five compartments: unvaccinated susceptible individuals (S); individuals who have received at least one dose of the two-dose vaccine but have yet to develop protection (V); individuals who has received both vaccine doses but failed to get protection (U); infectious individuals (I); and immune individuals (either recovered from the infection or protected by the vaccine) (R) (Supplementary Fig. 1). Note that the infectious compartment (I) includes both asymptomatic and symptomatic infections as no statistical difference in transmissibility was found between them[22] and we are not explicitly simulating interventions that act differently on those two groups. Nonetheless, a sensitivity analysis where we assume that the infectiousness of asymptomatic individuals relative to symptomatic ones is 0.5 has been conducted.

The transmission depends on contacts between susceptible individuals (i.e., those in compartments S, V, and U) and infectious individuals (I) and the risk of infection given a contact (β). Contacts are modeled through the use of a contact matrix $C_{i,j}$ representing the mean number of contacts that an individual in age group $i$ has with individuals in age group $j$. Contact patterns are estimated by relying on 2017/2018 survey data for Shanghai[23]. We use bootstrap (sample with replacement where the sampling weights are given by the distributions of age groups in China) to estimate mixing patterns at the country level. We estimate the contact patterns using the package 'socialmixr' in R version 4.0.3[24]. The resulting contact matrix is shown in Supplementary Fig. 21a. We refer to this contact matrix as of "pre-pandemic" contact matrix; a sensitivity analysis considering mixing patterns in China estimated in March 2020[15] (after the lockdown has ended) is presented as well (Supplementary Fig. 21b). The number of individuals in each age group is taken from the world population prospects 2019 for China[25] and denoted with $N_i$.

We also consider age-specific susceptibility to infection $s_i$ as estimated in reference[22]. For simplicity, we denote the contact matrix multiplied by the susceptibility to infection by age as $C_{i,j}^s = s_i C_{i,j}$. The mean generation time is set at 5.5 days[22] (and the rate of transition, which is the inverse of the generation time, is denoted with γ).

We consider a delay of $1/w$ days between the administration of the first vaccine dose and protection[26]; $e_i$ represents the vaccine efficacy for age group $i$. We denote by $v_i(t)$ the allocation decision variables for age group $i$ on the day $t$. In other words, $v_i(t)$ represents the number of individuals who receive the first vaccine at time step $t$. We consider the "all-or-nothing" mechanism, where vaccines are assumed to provide full immunity for $e_i$ proportions of vaccinated individuals and no immunity to the remaining $1-e_i$.

The model is represented by the following system of differential equations:

$$\frac{dS_i(t)}{dt} = -v_i(t) - S_i(t) \cdot \beta \sum_{j=1}^{J} C_{i,j}^s \frac{I_j(t)}{N_j}$$

$$\frac{dV_i(t)}{dt} = v_i(t) - wV_i(t) - V_i(t) \cdot \beta \sum_{j=1}^{J} C_{i,j}^s \frac{I_j(t)}{N_j}$$

$$\frac{dU_i(t)}{dt} = (1-e_i) \cdot wV_i(t) - U_i(t) \cdot \beta \sum_{j=1}^{J} C_{i,j}^s \frac{I_j(t)}{N_j} \qquad (1)$$

$$\frac{dI_i(t)}{dt} = (S_i(t) + V_i(t) + U_i(t)) \cdot \beta \sum_{j=1}^{J} C_{i,j}^s \frac{I_j(t)}{N_j} - \gamma I_i(t)$$

$$\frac{dR_i(t)}{dt} = \gamma I_i(t) + e_i \cdot wV_i(t).$$

Note that the model is run in conjunction with the constrains (Eq. (5)) that guarantee that $v_i(t) \le S_i(t)$ for all $t$.

We consider five types of epidemiological interest: infections, symptomatic cases, hospitalizations, ICUs, and deaths, that we refer to as "risk" metrics. Only the number of infections is directly given by the model. To calculate the other four quantities, we relied on a post-hoc analysis considering age-specific risk factors according to the literature[27–29]; parameter values are reported in Supplementary Table 2.

We then use the model to minimize the total risk incidence, namely the sum of daily new risk incidences over $T$ days, for each of the five risk metrics. Mathematically, this can be written as

$$\min \sum_{t=0}^{T-1} \sum_{i=1}^{J} r_i^{type} \cdot (I_i(t+1) - (1-\gamma)I_i(t)) \qquad (2)$$

where $r_i^{type}$, type ∈{symp, hosp, ICU, death, infec}, represents that, among infections, the risks of symptomatic infections, requiring hospitalizations, ICU admissions and deaths in age group $i$, with $r_i^{infec}$ simply set to 1 for all age groups.

**Baseline scenario**. We assume the vaccine to be efficacious in preventing the infection (with no efficacy in preventing disease given infection). The efficacy of the vaccine currently administrated in China was estimated to be 80% for the age group 15–59 years[30]; for the other age groups we used a 25% reduction[31,32] (i.e., 0.75×80%). We assume that the first does not confer protection[26,31]. Vaccinated people may get protection after the second dose take effect, 14 days after its administration plus 21 days after the first dose (i.e., $w = 1/(14+21)$ day$^{-1}$)[26]. We also assume the vaccines has a long-term immunity (i.e., longer than the full study period T, T = 400, days).

The transmission rate β is calculated using the next-generation matrices approach[33],

$$\beta = \frac{R \cdot \gamma}{\max \{\text{eigenvalues}(C_{i,j}^s)\}} \qquad (3)$$

As baseline value, we assume the reproduction number R to be 1.5, to account for the effect of NPIs. Other values are explored as sensitivity analysis.

All the parameters in the study are calibrated based on state-of-the-art knowledge of COVID-19 epidemiology and vaccination in China. Supplementary Table 1 summarizes the parameters.

We consider the vaccination campaign to start at the same time of an epidemic outbreak, which is initialized with one infectious individual in each age group. We have conducted an extensive sensitivity analysis on the delay between the epidemic

onset and the start of the vaccination campaign. We have also included a scenario where the campaigns end before the epidemic onset. Since Wuhan was the only location in China that has experienced prolonged local transmission of SARS-CoV-2, we assume a fully susceptible population at the beginning of the simulation.

**Vaccination parameters**. We consider 2.0 million first doses administrated per day (0.14% rollout speed), according to the number of daily doses administered in China from Mar 1, 2021 to May 8, 2021[9]. Each age group is stratified into two categories: essential workers (tier 1) and other individuals (tier 2). Essential workers, comprising 3.28% of the full populations (Supplementary Table 2), include healthcare workers (either front-line or not) and workers in the following sectors: law enforcement and security, nursing home and social welfare institutes, community, energy, food and transportation[34]. Vaccines are administered to essential workers first[34,35]. Then, we investigate optimal vaccine allocation strategies to the general population. In the baseline analysis, vaccines are administered to susceptible individuals only; we also conducted two sensitivity analyses where this assumption is relaxed.

**Optimization methods**. We explore a two-step optimization strategy to solve model (2). First, we solve the problem using the myopic strategy where we minimize daily outcomes iteratively for 400 days. Second, we use the dynamics from the myopic strategy to construct an approximation counterpart and solve it over the full period. Although the myopic strategy does not attempt to minimize the total outcomes over the full period, they greatly reduce the total outcomes, performing close to the optimal solutions in similar problems[36].

Due to the tier constraint in vaccine allocation, we break down the allocation for each age group $v_i(t)$ into two variables, as

$$v_i(t) = \sum_{k=1}^{2} v_{i,k}(t), \qquad (4)$$

for each $i = 1, \ldots, J$.

The optimization method is divided into the two following steps.

Step 1: Myopic optimal strategy. At the beginning of each day, we optimize vaccine allocation to minimize the risk incidences on the day:

$$\min \sum_{i=1}^{J} r_i^{type}(I_i(t+1) - (1-\gamma)I_i(t)), \qquad (5a)$$

$$\text{s.t.} \sum_{k=1}^{2}\sum_{i=1}^{J} v_{i,k}(t) \le c, \qquad (5b)$$

$$0 \le \sum_{k=1}^{2} v_{i,k}(t) \le S_i(t), i = 1, \ldots, J, \qquad (5c)$$

$$W_{i,k}(t) = W_{i,k}(t-1) + v_{i,k}(t-1), i = 1, \ldots, J, k = 1, 2, \qquad (5d)$$

$$W_{i,k}(t) \le N_{i,k}d_{i,k}, i = 1, \ldots, J, k = 1, 2, \qquad (5e)$$

$$\sum_{i=1}^{J} v_{i,2}(t) \le b_i(t)M, \qquad (5f)$$

$$\sum_{i=1}^{J} v_{i,1}(t) \le (1 - b_i(t))(M + W_{i,1}(t) - N_{i,1}a_{i,1}), \qquad (5g)$$

$$v_{i,1}(t) \ge c\left(\frac{N_{i,1}}{\sum_{j=1}^{J} N_{j,1}} - b_i(t)\right), i = 1, \ldots, J, \qquad (5h)$$

$$b_i \in \{0,1\}, S_i(t), I_i(t), V_i(t), U_i(t) \ge 0, i = 1, \ldots, J$$

where $r_i^{type}$, $type \in \{infec, symp, hosp, death, ICU\}$.

Objective function (5a) minimizes the total risk incidence across age groups on that day. For risk measures other than infections such as death, the myopic optimization accounts for the number of individuals that will eventually die among those who were infected at the time $t$. Vaccine allocation across tiers and age groups cannot exceed the daily capacity (constraint 5b) or the unvaccinated susceptible populations (constraint 5c). Constraint (5d) tracks the cumulative number of administrated vaccines for each age group within each tier, denoted by $W_{i,k}(t)$, and constraint (5e) limits the cumulative number of administrated vaccines, where $d_{i,k}$ represents the maximum coverage for age group $i$ within tier $k$ and $N_{i,k}$ the population size for age group $i$ within tier $k$. Constraints (5f) and (5g) guarantee that people in the first tier are vaccinated before individuals in the second tier. $M$ is a large number, with $M = c + N_{j,1}$ in the implementation. Constraint (5h) ensures that vaccines are uniformly administrated among age groups in the first tier.

Model (5) is a linear optimization problem with box constraints. Geometrically, the optimal solutions rest at the corners of the polyhedron comprised by the box constraints. In the objective function, we consider that the number of vaccinated individuals $v_i$ instantly moves to compartment $V_i$, while $(1 - e_i) \cdot wv_i$ are moved to compartment $U_i$ and $e_i \cdot wv_i$ to compartment $R_i$. Note that in this case, the

objective function (5a) is equivalent to $\sum_{i=1}^{J} r_i^{type}(S_i(t) + V_i(t) + U_i(t) - e_i w \sum_{k=1}^{2} v_{i,k}(t))\beta \sum_{j=1}^{J} C_{i,j}^s \frac{I_j(t)}{N_j}$. The myopic optimization thus determines the prioritization based on the modeled risk metrics. Namely, the myopic policy gives the highest priority to the age group that has the largest risk at the time $t$. If the number of unvaccinated susceptible population is smaller than the vaccination capacity $c$, it then diverts to the age group that has the second largest risk. If there are vaccines left after satisfying the first two groups, it further diverts to the third group that has the third largest risk, and so forth. Moreover, because the myopic optimal strategy performs well and close to the optimal solution in similar problems[36], the final optimal solutions is expected to share similar properties to it.

The model was coded in Gurobi R interface with R version 4.0.3[24] and solved using Gurobi 9.10[37]. We simplify the implementation of the model by replacing the last three constraints on vaccine allocation to groups in tier 1 with a pre-allocation. Specifically, we first allocate vaccines uniformly to the age groups in tier 1, and then use the remaining vaccines, if there is any, to optimize the allocation to the age groups in tier 2.

Using aggregated $v_i(t)$, $i = 1, \ldots, J$, from the myopic solutions, we update all the states to get the states status on day $t + 1$. We solve the equations using lsoda ODE solver from the package 'deSolve'[38] in R version 4.0.3[24]. This iteration of optimization-updating procedure was repeated from day 0 to day $T$, generating the myopic solutions for the full period.

Step 2: Approximated optimization. Using the myopic solutions from the Step 1, we explore the approximation strategy and minimize the total outcomes of the targeted risk metric. The full model is:

$$\min \sum_{t=0}^{T-1}\sum_{i=1}^{J} r_i^{type}(I_i(t+1) - (1-\gamma)I_i(t)) \qquad (6a)$$

$$\text{s.t.} \sum_{k=1}^{2}\sum_{i=1}^{J} v_{i,k}(t) \le c, t = 0, \ldots, T, \qquad (6b)$$

$$0 \le \sum_{k=1}^{2} v_{i,k}(t) \le S_i(t), i = 1, \ldots, J, t = 0, \ldots, T, \qquad (6c)$$

$$W_{i,k}(t+1) = W_{i,k}(t) + v_{i,k}(t), i = 1, \ldots, J, k = 1, 2, t = 0, \ldots, T, \qquad (6d)$$

$$W_{i,k}(t) \le N_{i,k}d_{i,k}, i = 1, \ldots, J, k = 1, 2, t = 0, \ldots, T, \qquad (6e)$$

$$\sum_{i=1}^{J} v_{i,2}(t) \le b_i(t)M, t = 0, \ldots, T, \qquad (6f)$$

$$\sum_{i=1}^{J} v_{i,1}(t) \le (1 - b_i(t))(M + W_{i,1}(t) - N_{i,1}a_{i,1}), t = 0, \ldots, T, \qquad (6g)$$

$$v_{i,1}(t) \ge c\left(\frac{N_{i,1}}{\sum_{j=1}^{J} N_{j,1}} - b_i(t)\right), i = 1, \ldots, J, t = 0, \ldots, T, \qquad (6h)$$

$$\left|\sum_{j=1}^{J} C_{i,j}^s \frac{I_j(t)}{N_j} - \sum_{j=1}^{J} C_{i,j}^s \frac{I_j^{\text{Myopic}(t)}}{N_j}\right| \le \epsilon, t = 0, \ldots, T, \qquad (6i)$$

Eq. (1) with $I_j(t)$ replaced by $I_j^{\text{Myopic}(t)}$, $t = 0, \ldots, T$,

$$b_i \in \{0,1\}, S_i(t), I_i(t), V_i(t), U_i(t) \ge 0, i = 1, \ldots, J, t = 0, \ldots, T.$$

Constraint (6i) bounds the error of approximation. $\varepsilon$ is a small number to control the approximation errors.

To choose the candidate myopic optimal solutions (denoted with $I_j^{Myopic}(t)$) for constructing the approximation counterpart in the second step, we use the "try and error" method. We found that the myopic solution minimizing the symptomatic cases perform relatively well on the other four risk metrics. We therefore use the myopic solutions from the scenario to construct the approximation counterparts where we set $d_{i,k} = 1, i = 1, \ldots, J, k = 1, 2$, in (5e) and (6e). The model was coded in Gurobi Python interface with Python 3.9.0[39], and solved using Gurobi 9.10[37].

**Sensitivity analyses**. We designed a variety of sensitivity analyses to evaluate the robustness of optimal prioritization strategies and to estimate their benefits.

*Vaccination capacity.* We explore different levels of vaccination capacity $c = 1.0$, 1.5, 2.0, 2.5, 3.0, 3.5 million first doses per day. Moreover, as vaccination capacity may increase over time, we consider a scenario where the vaccination capacity linearly increases from 1.5 million first doses on the first day to 2.5 million first doses at day 400.

*Reproduction number.* We explore different levels of $R = 1.25, 1.5, 1.75, 2.0$.

*Vaccine efficacy.* While more evidence about vaccine efficacy is collected, we test alternative levels of vaccine efficacy: $e_i = 0.6, 0.7, 0.9$, for $i = 4, …., 12$ and $e_i = 0.6 \times 0.75$, $0.7 \times 0.75$ or $0.9 \times 0.75$ for $i \leq 3$ or $i \geq 14$.

Moreover, we conducted an analysis where we consider the vaccine efficacy to be homogenous by age; in particular, we assume $e_i = 0.8$, for $i = 1, …., 17$.

*Vaccine hesitancy.* People may be hesitant to accept a COVID-19 vaccine for a variety of reasons[40]. To model vaccine hesitancy, we let $d_{i,k}$ in (6e) to be equivalent to the vaccine acceptancy, $a_{i,k}$. We estimate $a_{i,k}$ by using the data from a large-scale telephone survey conducted in June 2020 (unpublished results obtained by author H.Y. and his team). The potential acceptance of a COVID-19 vaccine within the first tier was estimated to be 96%. The estimated potential acceptance in the second tier is relatively stable by age, ranging from 78 to 89% (see Supplementary Table 2). Furthermore, we conduct an analysis assuming 61% as vaccine acceptance for all age groups in the general population, as estimated in reference[41] where a global-scale survey was conducted.

*Age groups eligible for vaccination.* As for general populations, the vaccines are mainly administered to people aged 18–59 years, we test the scenario where people under 20 and over 60 are excluded in vaccinations.

*Differential timing of vaccination campaign relative to the epidemic.* We conduct a set of analyses where the vaccination program starts between 90 days before to 30 days after the epidemic onset ($\delta = −90, −60, −30, −15, 15$, and 30 days). Note that in all the analyzed scenarios, vaccines are allocated to essential workers first; this roughly corresponds to the first 24 days of the simulation when considering 2 million first doses administered per day.

*Differential vaccine efficacy in preventing infection vs. disease.* In the baseline analysis, we have assumed the vaccine to be efficacious in preventing the infection with no additional effect in preventing the disease. Here we conduct an alternative analysis with 60% efficacy for preventing the infection, and four efficacy levels $e_i^{\text{Symp|Infection}}$ in preventing the disease given the infection, namely: 0%, 25%, 50% and 75%, or equivalently four efficacy levels $e_i^{\text{Symp}}$ in preventing the disease: 60%, 70%, 80% and 90%, where,

$$e_i^{\text{Symp}} = e_i + (1 - e_i) \times e_i^{\text{Symp|Infection}}$$

Specifically, we set $e_i = 0.6$, $e_i^{\text{Symp|Infection}} = 0, 0.25, 0.50$ or $0.75$, for $i = 4, …., 12$, and $e_i = 0.6 \times 0.75$, $e_i^{\text{Symp|Infection}} = 0 \times 0.75$, $0.25 \times 0.75$, $0.50 \times 0.75$ or $0.75 \times 0.75$ for $i \leq 3$ or $i \geq 14$. This feature is incorporated into the model by defining new risk of disease after vaccination, $r_i^{\text{Vac\_symp}}$, by adjusting the risk of developing symptoms as follows:

$$r_i^{\text{Vac\_Symp}} = r_i^{\text{Symp}} \times (1 - e_i^{\text{Symp|Infection}}),$$

Following the same arguments used in the baseline scenario (see "Methods"), we update the risk of requiring hospitalization, ICU admission, and death as well ($r_i^{\text{Vac\_hosp}}$, $r_i^{\text{Vac\_ICU}}$, $r_i^{\text{Vac\_death}}$).

In addition, we conduct an extreme scenario with 0% efficacy for preventing the infection, but 80% in preventing the disease given infections, so that the final efficacy for preventing the disease is the same as that in baseline. Specifically, we set $e_i = 0$, for $i = 1, …., 17$, and $e_i^{\text{Symp|Infection}} = 0.8$ for $i = 4, …., 12$, $e_i^{\text{Symp|Infection}} = 0.8 \times 0.75$ for $i \leq 3$ or $i \geq 14$. We do not consider minimizing infections in this analysis because of the vaccine has 0% efficacy in preventing the infection.

*"Leaky" vaccine.* We also consider a variation of model (1) where, for each age group $i$, instead of assuming a fully protective vaccine for $e_i$ proportion of vaccinated individuals, we consider that vaccination induces an $e_i$ reduction of the risk of infection ("leaky" vaccine) for all vaccinated individuals. This alternative model is represented by the following system of equations:

$$\frac{dS_i(t)}{dt} = -v_i(t) - S_i(t) \cdot \beta \sum_{j=1}^{J} C_{i,j}^s \frac{I_j(t)}{N_j}$$

$$\frac{dV_i(t)}{dt} = v_i(t) - wV_i(t) - V_i(t) \cdot \beta \sum_{j=1}^{J} C_{i,j}^s \frac{I_j(t)}{N_j}$$

$$\frac{dU_i(t)}{dt} = wV_i(t) - U_i(t) \cdot (1 - e_i) \cdot \beta \sum_{j=1}^{J} C_{i,j}^s \frac{I_j(t)}{N_j} \tag{7}$$

$$\frac{dI_i(t)}{dt} = (S_i(t) + V_i(t) + U_i(t) \cdot (1 - e_i)) \beta \sum_{j=1}^{J} C_{i,j}^s \frac{I_j(t)}{N_j} - \gamma I_i(t)$$

$$\frac{dR_i(t)}{dt} = \gamma I_i(t).$$

The full analysis conducted for the baseline scenario (all-or-nothing vaccine, $R = 1.5$) is repeated for the "leaky vaccine" model.

*Age-mixing pattern.* The contact matrix used in the baseline analysis refers to Shanghai, China and was estimated before the COVID-19 pandemic[23]. This contact matrix is here referred as to "pre-pandemic" contact matrix. As sensitivity analysis, we considered an alternative contact matrix ("pandemic" contact matrix) that was estimated from contact diaries collected in Shanghai in March 2020, at a time when the lockdown was over, but several non-pharmacological interventions were still in place[15]. The "pandemic" contact matrix highly differs from that used in the main analysis, showing a much less assortative pattern by age (Supplementary Fig. 21).

*Differential infectiousness of asymptomatic and symptomatic individuals.* In the main analysis, we assume that the infectiousness of asymptomatic cases and symptomatic cases are identical[22]. Here we consider a scenario where the infectiousness of asymptomatic individuals relative to symptomatic ones is set to 50%.

We denote $\beta^{\text{Symp}}$ the transmission rate of symptomatic cases and the transmission rate of asymptomatic cases, therefore, $0.5 \times \beta^{\text{Symp}}$. We also calculate $\beta^{\text{Symp}}$ through the next generation matrix, Eq. (3), with $C_{i,j}^s$ replaced by $C_{i,j}^s r_j^{\text{Symp}} + 0.5 C_{i,j}^s (1 - r_j^{\text{Symp}})$. The adjustment is made to maintain the same production number $R$ as in baseline. We note that

$$\beta^{\text{Symp}} \sum_{j=1}^{J} C_{i,j} \frac{I_j(t) \cdot r_j^{\text{Symp}}}{N_j} + 0.5 \beta^{\text{Symp}} \sum_{j=1}^{J} C_{i,j} \frac{I_j(t) \cdot (1 - r_j^{\text{Symp}})}{N_j}$$

$$= \beta^{\text{Symp}} \sum_{j=1}^{J} (C_{i,j}^s \cdot r_j^{\text{Symp}} + 0.5 C_{i,j}^s (1 - r_j^{\text{Symp}})) \frac{I_j(t)}{N_j}.$$

*Individuals eligible for vaccination.* In the main analysis, we assume that vaccines are allocated to susceptible individuals only. Here we consider two alternative scenarios where: (i) the vaccine is offered to any individual, regardless of her/his history of infection, and (ii) in addition to susceptible individuals, individuals who were infected become eligible for vaccination 180 days after they were originally infected[42].

Specifically, we consider that a proportion $p_i(t)$ of vaccines is assigned to susceptible individuals, with

$$p_i(t) = \frac{S_i(t)}{S_i(t) + \sum_{t=0}^{t-T'} (I_i(t+1) - (1 - \gamma) I_i(t)) \cdot (1 - r_i^{\text{death}}) - X_i(t)},$$

$$X_i(t) = X_i(t-1) + v_i(t-1) \times (1 - p_i(t-1)),$$

where $X_i(t)$ is added to track the vaccines allocated to infected people, $X_i(0) = 1$, $i = 1, …, J$.

Moreover, we replace $v_i(t)$ in Eqs. (1), (5) and (6) by $\bar{v}_i(t)$, with $\bar{v}_i(t) = v_i(t) \times p_i(t)$. We consider two cases: $T' = 180$[42] and $T' = 1$ respectively.

*Uncertainty in reported cases.* As sensitivity analysis, we consider a scenario accounting for the uncertainty in the reporting and reporting delay of cases. In particular, we assume that observed number of daily infections $\tilde{I}_i(t)$ follows Poisson distribution, with $\tilde{I}_i(t) \sim \text{Poisson}(I_i(t - \Delta t))$. We conducted two scenario analyses: (i) $\Delta t = 0$, i.e., no delays and, on each day, the expectation of the number of observed infections is the number of infections; (ii) $\Delta t = 11$[22], i.e., we consider a lag of 11 days between the infection and reporting and, on each day, the expectation of the number of observed infections is the number of infections 11 days before.

Specifically, to account for the uncertainty on the reporting, Eq. (2) is replaced by

$$\min \mathbf{E} \left( \sum_{t=0}^{T-1} \sum_{i=1}^{J} r_i^{\text{type}} \cdot (\tilde{I}_i(t+1) - (1 - \gamma) \tilde{I}_i(t)) \right) \tag{8}$$

where the expectation $\mathbf{E}(.)$ is taken with respect to the random noise associated with the uncertainties.

To largely account for the uncertainty in the optimization, we adjust the "Step 1: Myopic optimal strategy", with the $\tilde{I}_i(t)$ used in the daily objective and used objective functions (8) in the "Step 2: Approximated optimization".

*Vaccination campaign completes before epidemic onset.* Here we consider a scenario where the vaccination campaign completes before the epidemic onset. The optimization problem thus becomes a static optimization problem, where the allocation decision variables are reduced into $v_{i,k}(0)$. The vaccination campaign lasts $T'$ days and completes prior to the epidemic. Considering the 35-day delay between the administration of the first dose and protection, we test $T' = 100 + 35, 150 + 35, 400 + 35$ respectively. The equations regulating this analysis are:

$$\min \sum_{t=0}^{T-1} \sum_{i=1}^{J} r_i^{\text{type}} (I_i(t+1) - (1 - \gamma) I_i(t)) \tag{9a}$$

$$\text{s.t.} \sum_{k=1}^{2} \sum_{i=1}^{J} v_{i,k}(0) \leq cT', \tag{9b}$$

$$\sum_{k=1}^{2} v_{i,k}(0) \geq N_{i,1}, i = 1, … , J, \tag{9c}$$

$$\text{Equations (1) with } S_i(0) = N_i - I_i(0) - \sum_{k=1}^{2} v_{i,k}(0), i = 1, \dots, J, \quad (9d)$$

$$U_i(0) = (1 - e_i) \sum_{k=1}^{2} v_{i,k}(0), i = 1, \dots, J, \quad (9e)$$

$$R_i(0) = e_i \sum_{k=1}^{2} v_{i,k}(0), i = 1, \dots, J, \quad (9f)$$

$$S_i(t), I_i(t), U_i(t) \geq 0, V_i(t) = 0, i = 1, \dots, J, t = 0, \dots, T,$$
$$v_i(t) = 0, i = 1, \dots, J, t = 1, \dots, T.$$

**Reporting summary**. Further information on research design is available in the Nature Research Reporting Summary linked to this article.

## Data availability

All the data used in the study were detailed in Methods and provided in Supplementary Information. The data generated in this study are provided in the Supplementary Information/Source Data file. Source data are provided with this paper.

## Code availability

The code for the model used in this study is openly available at https://github.com/ShashaHan-collab/DynamicVaccineAllocationMod (https://doi.org/10.5281/zenodo.5090368).

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

## Acknowledgements

This study was supported by funding from the National Science Fund for Distinguished Young Scholars (No. 81525023), National Key Research and Development Program of China, and the grants from National Science Fund of China (No. 82041023, No. 81773546).

## Author contributions

H.Y. designed the study. M.A., X.-H.Z. and H.Y. supervised the work. S.H. developed the model. S.H. and J.C. conducted the research. J.C., J.Z., Q.W., W.Z., H.S. and S.H. collected data. S.H., J.C., J.Y., M.A., X.-H.Z. and H.Y. interpreted the findings. S.H., J.C. and M.A. wrote the manuscript. J.Y., X.-H.Z. and H.Y. commented on and revised the manuscript. All authors approved the final manuscript as submitted.

## Competing interests

M.A. has received research funding from Seqirus. H.Y. has received research funding from Sanofi Pasteur, GlaxoSmithKline, Yichang HEC Changjiang Pharmaceutical Company, and Shanghai Roche Pharmaceutical Company. None of those research funding is related to COVID-19. All other authors report no competing interests.
