## [Peer Review File · Nature Communications]

Comments to Authors

Manuscript ID: NCOMMS-21-07406-T

I have reviewed this manuscript from the perspective of a mathematical epidemiologist who has experience in infectious disease modelling.

In the study, the authors present a mechanistic model of SARS-CoV-2 transmission to explore vaccine prioritisation strategies that reactively adapt to the epidemiological situation. A data-driven approach was used (with data pertaining to China) and a collection of risk metrics assessed (infections, cases, hospital admissions, ICU admissions, deaths).

The study found that vaccination programmes that reactively adapt to the epidemiological situation are capable of simultaneously achieving different objectives. Furthermore, those population groups identified as highest priority under the baseline parameter scenario were broadly maintained under sensitivity to vaccine efficacy, differential vaccine efficacy in preventing infection vs. disease, vaccine hesitancy, and SARS-CoV-2 transmissibility. The work contributes to a topical theme of how limited vaccine supplies would be best allocated (in the context of SARS-CoV-2).

Major comments

I outline below what are, in my opinion, items to undergo further consideration.

A constant challenge is that the number of new cases is not precisely known. Mathematical models can be used to provide a predicted range of estimated possibilities on case load, given the available evidence. Estimations of incidence commonly require models first being calibrated to available data streams measuring public health measurable quantities (e.g. mortality, hospitalisations, positive PCR tests). Yet, these data streams themselves have reporting lags. Though community surveillance projects could also provide insights on incidence and prevalence, these results are not instantaneously producible either. Practically, there will be uncertainty/an estimated range considered plausible for the number of infected individuals on a given day. Given these constraints, and with a view of practical operational use of this methodology in terms of policy, performing an analysis where there is uncertainty in the number of presumed infectious incorporated would be advantageous.

What would be the implications if the assumption of an asymptomatic case having the infectiousness as symptomatic case were relaxed? These caveats warrant explanation in the discussion.

From the model equations and the text throughout the methods, am I interpreting correctly that all vaccinations are administered to unvaccinated susceptibles only? If so, this will need recognising as a model assumption; what may be the impact if those who have previously been infected were equally likely to receive a vaccine? The extent this could alter the outcome of the optimisation could be looked into.

As part of the overview of relevant literature, there is a study from Bubar *et al* (first posted as a preprint on medRxiv in late June 2020 and recently published in Science) that in part analyses sensitivity to rollout (Figure S6 a particular example from that work). It would be interesting to have a summary of how these analyses coincide and where outcomes may differ.

Reference: Model-informed COVID-19 vaccine prioritization strategies by age and serostatus BY KATE M. BUBAR, KYLE REINHOLT, STEPHEN M. KISSLER, MARC LIPSITCH, SARAH COBEY, YONATAN H. GRAD, DANIEL B. LARREMORE SCIENCE | 26 FEB 2021: 916-921.

Minor comments

Please find below additional line-by-line comments, which I think would help aid the quality and clarity of the manuscript.

Introduction

Reference 3. Apologies, after looking at the article I could not find text relating to the statement. (though within that article there was a reference to the Financial Times article to not everyone having vaccine before 2024).

Results

L66: "we minimize the total incidences over 400 days." Why was the time horizon set at 400 days? Was that to correspond with the time elapsed until there was full vaccine coverage of those aged 18+?

L96: "Not only the identified priority orders". Amend to "Not only are the identified priority orders".

L102-103: "vaccination increases with vaccine supply 2.0 million courses per day (0.14% rollout speed) is reached". This is unclear. Please rephrase.

L124: "R=1.5 and all-or-nothing vaccine". Amend to "R=1.5 and an all-or-nothing vaccine".

Additionally, it would be helpful for the assumption of essential workers vaccinated first to also be stated in the results. How many & what fraction of the overall population comprises essential workers? A linking paragraph/scene-setting paragraph would be beneficial.

Discussion

L130: "population segments as epidemic unfolds". Amend to "population segments as an epidemic unfolds".

L131: "as comparison to" amend to "as a comparison to".

L170-172: The contextual information provided here on the eligible age groups for vaccine in China I consider as key information that should also be stated in the Introduction.

L180: "'Despite we recognize" does not parse. "Though we recognize" could be a suitable replacement.

L194: "framework presented here general enough" change to "framework presented here is general enough".

Methods

P17, Paragraph 2, Line 1: Replace "maxim" with "maximum".

P21&P,23 Eqns (5h,6h): Should the numerator of the fraction have subscript "i,1" rather than "j,1"?

P24, Vaccine hesitancy: "we let $d_{i,k}$ in (6e)". Does that also apply to equation (5e)?"

P24, Vaccine hesitancy: “The potential acceptance of a COVID-19 vaccine within the first tier was estimated to be 96%.” Is there a reference for this estimate?

Figures & Tables

For line plots, to aid interpretability for the reader and more simply distinguish between different lines please consider varying the line styles (and marker types).

Reviewer #2 (Remarks to the Author):

This study addresses a very important question, of how vaccination strategy to COVID-19 should be optimised, in order to maximise the impact of a limited vaccine stockpile. The authors show how a time-varying strategy, that focuses on different population groups over time in a specified way, can be more impactful than random vaccination. I found the study very interesting, but have a couple of concerns on methodology and clarity, as follows.

Mechanisms of vaccine efficacy: One key assumption in the modelling is that the vaccine efficacy acts through preventing infection. This isn't necessarily true, as trial-based efficacy estimates tell us about effect on symptomatic illness: it's not always clear whether this effect arises from blocking infection, or rather allowing infection and moderating disease.

The authors have performed a sensitivity analysis when assuming the infection-preventing efficacy is augmented with disease-reducing efficacy, but I don't think this is the key analysis that's needed. Instead, the authors should compare two scenarios with equal efficacy: one where the vaccine reduces infection but not severity of symptoms, and the other where the vaccine has no effect on infection, but reduces severity of symptoms. Theoretically, the latter scenario could well cause some important changes in the optimal sequence of vaccination, since it won't be as effective in reducing transmission (in principle, resulting in increasing priority towards the most vulnerable groups).

Clarity on model specification: I found the transmission model a little hard to understand, and this could do with some adjustment/more careful explanation. For example, in the governing equations on p.17 of the supporting information, I understand that U denotes those who received vaccination but failed to get immunity, while V denotes those vaccinated who successfully gained immunity. (This wasn't immediately clear from the text, and could do with some clarification.)

The role of w is quite confusing in the equations: although it is defined as a rate in the text, it seems to be used as a proportion in the equations (with appearances of w and $1-w$). A better approach would have been to define an additional model compartment, intermediate to S and U/V, to represent those who have received vaccination but haven't yet completed the two-week period for immunity to come into effect.

Also confusing is the role of $v_i(t)$, the key variable governing the allocation decisions. In the equation for dV_i/dt , by the fact that $v_i(t)$ is directly multiplying beta, it is playing the role of a state variable denoting susceptible individuals (put another way, the equations would show some incidence of infection as long as $v_i > 0$, even if there were no susceptible individuals...which can't be right). Given the crucial driving role of $v_i(t)$ in the model dynamics, I strongly suggest the authors revisit this model design, as well as clarifying how exactly $v_i(t)$ should be interpreted.

Finally, the model equations do not distinguish asymptomatic/symptomatic infections, yet the results show outputs for minimising total symptomatic cases. Could the authors clarify? Similarly, the model does not include any terms for mortality, yet the results show optimal strategies for minimising deaths. Did the authors perform a post-hoc analysis, assuming that a certain proportion of incident cases would suffer mortality? If so, I suggest this would again need adjustment, or careful discussion, since we know that it can take up to a month for death to occur. Ignoring this timeline would skew the results, where simulated decisions for vaccine allocation are made on a daily basis.

Clarifying the scope of 'dynamical' strategies: It is helpful to distinguish a 'time-varying' vaccination

policy from a truly 'dynamic' one. The first is a strategy where vaccination coverage may change over time (e.g. shifting from one population group to another), but the policymaker knows in advance how and when these shifts will happen. The second is a strategy identifying certain epidemiologically-driven rules for how the policymaker will adapt their vaccination focus, in real time. Only these decision rules are known in advance, and not the actual specifics of coverage.

The abstract and introduction are written in such a way that it seems this study is addressing the second, but on a deeper reading, it seems that it is in fact addressing the first. For example, the main results in the paper specify certain schedules for vaccination coverage that would be optimum under different objectives (rather than specifying rules for deciding priority groups in real time). The overall message is essentially that targeting risk groups in a sequential way is better than uniform, static targeting: the main purpose of the study is to present optimal such sequences, but not to identify decision rules per se.

It would be helpful to make this more clear, to help the reader better understand the scope of this work. As one example, the summary refers to 'reactively adapting' the vaccination programme; I suggest the text should be clarified in these and other places, to clarify that the study does not present decision rules for real time, reactive vaccination.

Other comments:

The assumed contact matrix is likely to have a strong influence on the importance of different age-groups, when receiving an infection-blocking vaccine. It is good to see that this matrix is derived from local data; nonetheless, given the important underlying role of this matrix, there doesn't appear to be enough consideration of how it drives the results. It would be good to see a sensitivity analysis to the assumed matrix coefficients (It would probably be enough simply to adopt contact matrices from other, contrasting settings). Similarly, the authors make assumptions for age-specific efficacy, that should also be subjected to additional analysis, given their obvious importance for the response of different age groups, to vaccination.

Please also give more information on the relative timing of the vaccination campaign, in relation to the epidemic. I believe at the moment, this timing is implicit in the initial conditions chosen for the epidemic (i.e. assuming that the vaccination programme is also initiated at time zero, a greater number of initial infecteds would mean a later vaccination programme in relation to the epidemic, and vice versa). For example, how would the results change if the vaccination is so early that it is completed before any substantial epidemic occurs? In this case, presumably there would be no benefit in priority sequencing. I realise this may be a lot of additional work to incorporate in the model itself, but at least some discussion of the point (if not outright modelling) would be helpful.

Responses to Reviewer's comments

Referee #1 (Remarks to the Author):

I have reviewed this manuscript from the perspective of a mathematical epidemiologist who has experience in infectious disease modelling.

In the study, the authors present a mechanistic model of SARS-CoV-2 transmission to explore vaccine prioritisation strategies that reactively adapt to the epidemiological situation. A data-driven approach was used (with data pertaining to China) and a collection of risk metrics assessed (infections, cases, hospital admissions, ICU admissions, deaths).

The study found that vaccination programmes that reactively adapt to the epidemiological situation are capable of simultaneously achieving different objectives. Furthermore, those population groups identified as highest priority under the baseline parameter scenario were broadly maintained under sensitivity to vaccine efficacy, differential vaccine efficacy in preventing infection vs. disease, vaccine hesitancy, and SARS-CoV-2 transmissibility. The work contributes to a topical theme of how limited vaccine supplies would be best allocated (in the context of SARS-CoV-2).

Response: We would like to thank the reviewer for taking the time to assess our manuscript and the many useful comments.

Major comments

I outline below what are, in my opinion, items to undergo further consideration.

1. A constant challenge is that the number of new cases is not precisely known. Mathematical models can be used to provide a predicted range of estimated possibilities on case load, given the available evidence. Estimations of incidence commonly require models first being calibrated to available data streams measuring public health measurable quantities (e.g. mortality, hospitalisations, positive PCR tests). Yet, these data streams themselves have reporting lags. Though community surveillance projects could also provide insights on incidence and prevalence, these results are not instantaneously producible either. Practically, there will be uncertainty/an estimated range considered plausible for the number of infected individuals on a given day. Given these constraints, and with a view of practical operational use of this methodology in terms of policy, performing an analysis where there is uncertainty in the number of presumed infectious incorporated would be advantageous.

Response to comment #1: We would like to thank the reviewer for her/his useful input. We fully agree with the reviewer that, in practical situations, there will be uncertainty surrounding nearly any quantity of public health relevance (e.g., number of infections, number of deaths). As such, we added two new analyses considering two different possible sources of uncertainty: i) the uncertainty in the observed cases, and ii) the delay in reporting them.

Specifically, the two new analyses work as follows: in the optimization procedure, we consider the number of infected individuals of age i at time t to be

$$\tilde{I}_i(t) \sim \text{Poisson} (I_i(t - \Delta t))$$

where $I_i(t)$ is the number of infected individuals of age i at time t estimated by the model and Δt is the reporting delay.

The new analysis is described in the Methods section (Line 617-630). In sum, we found that the optimal strategies are consistent when accounting for uncertainty in reported cases and reporting lags and their effectiveness remain unaltered. The results of this new analysis are shown in Fig. 3 (panel 1), Supplementary Figure 14, and mentioned several times in the main text; for instance:

Line 193-195: *“The identified prioritization strategies and the estimated effectiveness of the vaccination campaign are consistent when accounting for uncertainty in case reporting and delays (Fig. 3l, Supplementary Fig. 14).”*

We would like to thank once again the reviewer for this comment as we believe that it allowed us to show the suitability of our methodology for operational use.

2. What would be the implications if the assumption of an asymptomatic case having the infectiousness as symptomatic case were relaxed? These caveats warrant explanation in the discussion.

Response to comment #2: The Reviewer rose an interesting point about heterogeneities in infectiousness between asymptomatic and symptomatic individuals. We have redesigned the model to account for this feature and run new analysis where we assume asymptomatic individuals to be half infectious than symptomatic individuals. We found that the priority categories broadly identified in this new scenario are consistent with those identified assuming an identical infectiousness. The results of this new analysis are reported in Fig. 3k, Supplementary Figs. 7 and 20b, and mentioned multiple times in the main analysis; for instance:

Line 91-99: “We estimate that, for each optimal strategy, the two age categories with the highest priority are broadly consistent under different hypotheses on vaccine supply, vaccine efficacy, differential vaccine efficacy in preventing infection and disease, vaccine hesitancy, timing of vaccination campaign relative to epidemic onset, SARS-CoV-2 transmissibility, lower infectiousness of asymptomatic individuals relative to symptomatic ones, and uncertainties in the observed cases (i.e., reporting lags). Not only are the identified priority orders identical (See Fig. 1, Supplementary Figs. 3, 6, 13 and 14), but also the associated coverages show only little variations in most cases (Supplementary Fig. 7).”

3. From the model equations and the text throughout the methods, am I interpreting correctly that all vaccinations are administered to unvaccinated susceptibles only? If so, this will need recognising as a model assumption; what may be the impact if those who have previously been infected were equally likely to receive a vaccine? The extent this could alter the outcome of the optimisation could be looked into.

Response to comment #3: The Reviewer is correct that all vaccinations are administered to unvaccinated susceptible populations and we apologize for the lack of clarity in the original manuscript text, which has now been amended (Methods Line 467-469: “Then, we investigate optimal vaccine allocation strategies to the general population. In the baseline analysis, vaccines are administered to susceptible individuals only;”).

To evaluate to what extent vaccinating susceptible individuals only could alter the outcome of our analysis, we have modified the model to simulate vaccination strategies where previously infected individuals are vaccinated regardless of whether they are still susceptible or not. We found that, while the prioritized categories slightly change, vaccinating also previously infected individuals do not dramatically alter the results of our baseline analysis (which assumes a fully susceptible population at the beginning of the outbreak). It is possible that such a result would be highly different in Western countries where at the start of the vaccination campaign a relatively large fraction of the population was already infected by SARS-CoV-2. The obtained results are reported in Fig. 3h, Supplementary Fig 21, are mentioned multiple times, and commented in the Discussion as follows:

Line 177-182: “Furthermore, we found that vaccinating individuals regardless of their infection history has a relatively small impact on the epidemic burden (Supplementary Figs. 21b), although the identified prioritization strategies differ from that in the baseline (Supplementary Figs. 21a). This finding could be associated with the vaccination capacity of the Chinese health system and the considered value of the reproduction number (i.e., 1.5) and it thus not necessarily be the case in other contexts¹⁷.”

4. As part of the overview of relevant literature, there is a study from Bubar et al (first posted as a preprint on medRxiv in late June 2020 and recently published in Science) that in part analyses sensitivity to rollout (Figure S6 a particular example from that work). It would be interesting to have a summary of how these analyses coincide and where outcomes may differ.

Reference: Model-informed COVID-19 vaccine prioritization strategies by age and serostatus BY KATE M. BUBAR, KYLE REINHOLT, STEPHEN M. KISSLER, MARC LIPSITCH, SARAH COBEY, YONATAN H. GRAD, DANIEL B. LARREMORE SCIENCE | 26 FEB 2021: 916-921.

Response to comment #4: As suggested, we have added the following paragraph to compare our results with those obtained by Bubar et al.:

Discussion Line 162-171: “We examined the connection between the daily vaccine supply (rollout speed) and transmissibility in determining the optimal strategies. In line with previous work¹⁶, we found that differences between prioritization strategies are negligible when the daily supply is high enough relative to the reproduction number R (Figs. 3a and 3i, Supplementary Figs. 2a and 2c). However, here we found that the optimal vaccination strategies consistently prioritize high risk population across varying scenarios including different rollout speeds, SARS-CoV-2 transmissibility, and vaccine mechanisms, although, depending on the scenario (e.g., Fig. 1, Supplementary Figs. 3a, 7 and 12c). This novel finding is likely associated with the methodology we used, which allowed us to consider strategies where the vaccination can target a new age group before the full coverage in the previous group is reached.”

Discussion Line 177-182: “Furthermore, we found that vaccinating individuals regardless of their infection history has a relatively small impact on the epidemic burden (Supplementary Figs. 21b), although the identified prioritization strategies differ from that in the baseline (Supplementary Figs. 21a). This finding could be associated with the vaccination capacity of the Chinese health system and the considered value of the reproduction number (i.e., 1.5) and it thus not necessarily be the case in other contexts¹⁷.”

Minor comments

Please find below additional line-by-line comments, which I think would help aid the quality and clarity of the manuscript.

We would like to thank the reviewer once again for taking the time to provide us with a set of detailed comments.

Introduction

5. Reference 3. Apologies, after looking at the article I could not find text relating to the statement. (though within that article there was a reference to the Financial Times article to not everyone having vaccine before 2024).

Response to comment #5: We thank the Reviewer for pointing it out. We have replaced reference 3 with the original report: *Coronavirus vaccines: expect delays Q1 global forecast 2021*. Available at <https://img.lalr.co/cms/2021/01/28193636/report-q1-global-forecast-2021-1.pdf> (2021).

Results

6. L66: “we minimize the total incidences over 400 days.” Why was the time horizon set at 400 days? Was that to correspond with the time elapsed until there was full vaccine coverage of those aged 18+?

Response to comment #6: We apologize for the lack of detail. 400 corresponds to the duration of an epidemic without vaccination in our baseline analysis. This has been clarified in the revised text:

Line 68-69: “The optimization is carried out over a time window of 400 days, roughly corresponding to the duration of a simulated epidemic with no vaccination.”

7. L96: “Not only the identified priority orders”. Amend to “Not only are the identified priority orders”.

Response to comment #7: Thank you; correction made.

8. L102-103: “vaccination increases with vaccine supply 2.0 million courses per day (0.14% rollout speed) is reached”. This is unclear. Please rephrase.

Response to comment #8: We apologize for the lack of clarity. The sentence has been removed in the revised version of the manuscript.

9. L124: “R=1.5 and all-or-nothing vaccine”. Amend to “R=1.5 and an all-or-nothing vaccine”. Additionally, it would be helpful for the assumption of essential workers vaccinated first to also be stated in the results. How many & what fraction of the overall population comprises essential workers? A linking paragraph/scene-setting paragraph would be beneficial.

Response to comment #9: Thank you; correction made. Moreover, we apologize for the lack of details in the Results. Essential workers correspond to about 3% of the Chinese population. We have revised the text as follows:

Results Line 62-63: “*Note that essential workers, corresponding to 3.28% of the total population (Supplementary Table 2), are vaccinated first.*”

Discussion

10. L130: “population segments as epidemic unfolds”. Amend to “population segments as an epidemic unfolds”.

Response to comment #10: Thank you; correction made.

11. L131: “as comparison to” amend to “as a comparison to”.

Response to comment #11: Thank you; correction made.

12. L170-172: The contextual information provided here on the eligible age groups for vaccine in China I consider as key information that should also be stated in the Introduction.

Response to comment #12: We thanks the Reviewer for pointing this out. We have amended the Introduction to include this information:

Line 45-48: “*A two-dose vaccination campaign has been started in China with vaccines administrated first to essential workers (3.28% of the full populations; Supplementary Table 2), then to the general populations of aged 18-59 years, and later extended to all 18+ individuals⁸. As of*

May 8, 2021 about 8% of general population has been vaccinated with two doses (317.5 million doses⁹).

13. L180: “Despite we recognize” does not parse. “Though we recognize” could be a suitable replacement.

Response to comment #13: Thank you; correction made.

14. L194: “framework presented here general enough” change to “framework presented here is general enough”.

Response to comment #14: Thank you; correction made.

Methods

15. P17, Paragraph 2, Line 1: Replace “maxim” with “maximum”.

Response to comment #15: Thank you; correction made.

16. P21&P,23 Eqns (5h,6h): Should the numerator of the fraction have subscript “i,1” rather than “j,1”?

Response to comment #16: The Reviewer is correct and we are grateful to her/him for catching this mistake. We have revised the text accordingly.

17. P24, Vaccine hesitancy: “we let $d_{i,k}$ in (6e)”. Does that also apply to equation (5e)?“

Response to comment #17: In the revised analysis, this does apply to equation (5e) as well. We modified the text as follows to clarify it:

Methods Line 522-523: “We therefore use the myopic solutions from the scenario to construct the approximation counterparts where we set $d_{i,k} = 1$, $i = 1, \dots, J$, $k = 1, 2$, in (5e) and (6e).”

18. P24, Vaccine hesitancy: “The potential acceptance of a COVID-19 vaccine within the first tier was estimated to be 96%.” Is there a reference for this estimate?

Response to comment #18: We apologize for the lack of detail. This estimate was obtained through a telephone survey conducted by author Hongjie Yu and his team at Fudan University. A manuscript on the topic is in preparation. The revised sentence now reads

Methods Line 544-547: “We estimate $a_{i,k}$ by using the data from a large-scale telephone survey conducted in June 2020 (unpublished results obtained by author H.Y. and his team). The potential acceptance of a COVID-19 vaccine within the first tier was estimated to be 96%.”

Figures & Tables

19. For line plots, to aid interpretability for the reader and more simply distinguish between different lines please consider varying the line styles (and marker types).

Response to comment #19: Thanks for this comment. We have modified the figures accordingly.

Referee #2 (Remarks to the Author):

This study addresses a very important question, of how vaccination strategy to COVID-19 should be optimised, in order to maximise the impact of a limited vaccine stockpile. The authors show how a time-varying strategy, that focuses on different population groups over time in a specified way, can be more impactful than random vaccination. I found the study very interesting, but have a couple of concerns on methodology and clarity, as follows.

Response: We would like to thank the Reviewer for taking the time to evaluate our study and for the many constructive comments she/he provided. We are glad that the Reviewer believes that our study is “very interesting” and “addresses a very important question” .

1. Mechanisms of vaccine efficacy: One key assumption in the modelling is that the vaccine efficacy acts through preventing infection. This isn't necessarily true, as trial-based efficacy estimates tell us about effect on symptomatic illness: it's not always clear whether this effect arises from blocking infection, or rather allowing infection and moderating disease.

The authors have performed a sensitivity analysis when assuming the infection-preventing efficacy is augmented with disease-reducing efficacy, but I don't think this is the key analysis that's needed. Instead, the authors should compare two scenarios with equal efficacy: one where the vaccine reduces infection but not severity of symptoms, and the other where the vaccine has no effect on infection, but reduces severity of symptoms. Theoretically, the latter scenario could well cause some important changes in the optimal sequence of vaccination, since it won't be as effective in reducing transmission (in principle, resulting in increasing priority towards the most vulnerable groups).

Response to comment #1: As suggested by the reviewer, we have added a new analysis to compare the two scenarios with equal vaccine efficacy, but where the vaccine is efficacious either in preventing the infection (and consequently also the disease) or the disease only. The first scenario corresponds to our baseline scenario, where we assume that the vaccines have 80% (or $80\% \times 0.75$ depending on the age group) efficacy to prevent the infection (while it does not affect the probability of developing symptoms given the infection). In the second scenario we assume that the vaccines have 0% efficacy in preventing the infection and 80% (or $80\% \times 0.75$ depending on the age groups) in preventing the disease.

The new analysis is described in the Methods Line 559-576. In sum, we found that if the vaccine is efficacious against the disease only, prioritized strategies can still be effective in reducing the total number of patients requiring an ICU and the total number of deaths, while they have little/no effect in averting cases and infections. The obtained results are presented in Fig. 3f, Supplementary Fig. 4d and mentioned multiple times the main text.

Moreover, we have revised the original analysis to include a sensitivity where we consider the vaccine to have 60% efficacy in preventing the infection and 0%, 25%, 50%, or 75% efficacy in preventing the disease given the infection. In these four scenarios, the overall vaccine efficacy in preventing the disease thus become 60%, 70%, 80%, and 90%. The obtained results are commented in the main text as follows:

Line 187-192: “When vaccines have low efficacy (60%) in preventing the infection, the effectiveness of optimal prioritization strategies is stable (Fig. 3e), while the impact on reducing COVID-19 burden increases with the efficacy in preventing the disease (Supplementary Fig. 4a-4c). On the other hand, if the vaccine is protective against the disease only, prioritized strategies can still be effective in reducing the total number of patients requiring an ICU and the total number of deaths (Fig. 3f, Supplementary Figs. 4d and 11f).”

2. Clarity on model specification: I found the transmission model a little hard to understand, and this could do with some adjustment/more careful explanation. For example, in the governing equations on p.17 of the supporting information, I understand that U denotes those who received vaccination but failed to get immunity, while V denotes those vaccinated who successfully gained immunity. (This wasn't immediately clear from the text, and could do with some clarification.)

Response to comment #2: We apologize for lack of clarify. To respond to other comments by the same reviewer, we modified the equations regulating the model. Compartment V represents individuals who were vaccinated but they have yet to develop immunity. After the time from vaccination to protection, an individual can either move to compartment R (if she/he was successfully immunized) or move to compartment U, which contains individuals who failed to develop immunity. (We would like to remind that, in the main analysis, we are considering an “all-or-nothing” vaccine).

We have included a diagram (Supplementary Fig. 1, which is reported hereafter for reviewer convenience) to clarify the new system of equations and modified the main text as follows:

Line 393-398: “For each age group, the population is divided into five compartments: unvaccinated susceptible individuals (S); individuals who have received the vaccine but have yet to develop protection (V); individuals who has received the vaccine(s) but failed to get protection (U); infectious individuals (I); and immune individuals (either recovered from the infection or protected by the vaccine) (R) (Supplementary Fig. 1).”

Supplementary Fig. 1: Schematic representation of the transmission model. By omitting the dependency on age and time, model compartments are defined as follows: S represents unvaccinated susceptible individuals; V represents individuals who have received the vaccine but have yet to develop protection; U represents individuals who has received the vaccine(s) but fail to get protection; I represents infectious individuals; R represents individuals who are immune to the infection either due to recovery after natural infection or successful vaccination. The diagram applies to all-or-nothing model (Equation 1, used in the main analysis).

3. The role of w is quite confusing in the equations: although it is defined as a rate in the text, it seems to be used as a proportion in the equations (with appearances of w and $1-w$). A better approach would have been to define an additional model compartment, intermediate to S and U/V, to represent those who have received vaccination but haven't yet completed the two-week period for immunity to come into effect.

Response to comment #3: We apologize for the lack of clarity and thank the reviewer for the suggestion. As mentioned in the previous response, we have revised the system of equations. In the new system, V corresponds to the intermediate state described by the reviewer (i.e., individuals “who have received (the first dose of two-dose) vaccination but haven’t yet completed (the second dose and) the two-week period for immunity to come into effect”). Now, $1/w$ corresponds to the average delay between the administration of the first vaccine dose and protection:

Line 420-421: “We consider a delay of $1/w$ days between the administration of the first vaccine dose and protection²⁶”

4. Also confusing is the role of $v_i(t)$, the key variable governing the allocation decisions. In the equation for dV_i/dt , by the fact that $v_i(t)$ is directly multiplying beta, it is playing the role of a state variable denoting susceptible individuals (put another way, the equations would show some incidence of infection as long as $v_i > 0$, even if there were no susceptible individuals...which can’t be right). Given the crucial driving role of $v_i(t)$ in the model dynamics, I strongly suggest the authors revisit this model design, as well as clarifying how exactly $v_i(t)$ should be interpreted.

Response to comment #4: We apologize for the lack of clarity. $v_i(t)$ is the number of individuals who receive the first dose at time step t . For optimization purposes, this transition happens once at the beginning of each time step, instead of occurring on a continue scale. In other words, at the start of time step t , v_i individuals in compartment S_i are instantly moved to compartment V_i . As such, the force of infection in the first equation is applied to $S_i - v_i$ individuals, while the v_i individuals who received the first dose can be infected while in V_i compartment (second equation). We have modified the text as follows to clarify this concept:

Line 421-424: “We denote by $v_i(t)$ the allocation decision variables for age group i on the day t . In other words, $v_i(t)$ represents the number of individuals who receive the first dose at time step t . Specifically, at the beginning of each time step t , v_i individuals in compartment S_i are instantly moved to compartment V_i .”

Regarding the reviewer concern that the equations would show some incidence of infection as long as $v_i > 0$, even if there were no susceptible individuals, this is never the case in our system. In fact, constraints (5c) and (6c) prevent this to happen. Essentially, if there are no susceptible individuals for the group i on the day t , $v_i(t) = 0$. We have added the following sentence to clarify this point:

Line 428-429: “Note that the model is run in conjunction with the constrains (equation 5) that guarantee that $v_i(t) \leq S_i(t)$ for all t .”

5. Finally, the model equations do not distinguish asymptomatic/symptomatic infections, yet the results show outputs for minimising total symptomatic cases. Could the authors clarify? Similarly, the model does not include any terms for mortality, yet the results show optimal strategies for minimising deaths. Did the authors perform a post-hoc analysis, assuming that a certain proportion of incident cases would suffer mortality? If so, I suggest this would again need adjustment, or careful discussion, since we know that it can take up to a month for death to occur. Ignoring this timeline would skew the results, where simulated decisions for vaccine allocation are made on a daily basis.

Response to comment #5: The Reviewer is right. We have done post-hoc analyses to estimate the disease burden (similar to previous studies such as Bubar et al., Science, 2021). The reviewer is right that there is a wide delay between the date of infection and of death and we agree this is a limitation of our study done to ease the dynamical optimization. As suggested, we have added the following paragraph to comment on this limitation:

Line 430-434: “We consider five types of epidemiological interest: infections, symptomatic cases, hospitalizations, ICUs, and deaths, that we refer to as “risk” metrics. Only the number of infections is directly given by the model. To calculate the other four quantities, we relied on a post-hoc analysis considering age-specific risk factors according to the literature²⁷⁻²⁹; parameter values are reported in Supplementary Table 2.”

Moreover, we have performed a new sensitivity analysis where we consider an 11-day delay in the reporting of cases (see Methods, Lines 617-630). The results of this analysis are consistent to what obtained in the baseline analysis:

Line 193-195: “The identified prioritization strategies and the estimated effectiveness of the vaccination campaign are consistent when considering the uncertainty in reporting cases and reporting delays (Fig. 3l, Supplementary Fig. 14).”

6. Clarifying the scope of ‘dynamical’ strategies: It is helpful to distinguish a ‘time-varying’ vaccination policy from a truly ‘dynamic’ one. The first is a strategy where vaccination coverage may change over time (e.g. shifting from one population group to another), but the policymaker knows in advance how and when these shifts will happen. The second is a strategy identifying certain epidemiologically-driven rules for how the policymaker will adapt their vaccination focus, in real time. Only these decision rules are known in advance, and not the actual specifics of coverage.

The abstract and introduction are written in such a way that it seems this study is addressing the second, but on a deeper reading, it seems that it is in fact addressing the first. For example, the main results in the paper specify certain schedules for vaccination coverage that would be optimum under different objectives (rather than specifying rules for deciding priority groups in real time). The overall message is essentially that targeting risk groups in a sequential way is better than uniform, static targeting: the main purpose of the study is to present optimal such sequences, but not to identify decision rules per se.

It would be helpful to make this more clear, to help the reader better understand the scope of this work. As one example, the summary refers to ‘reactively adapting’ the vaccination programme; I suggest the text should be clarified in these and other places, to clarify that the study does not present decision rules for real time, reactive vaccination.

Response to comment #6: We apologize for the unclarity. As the Reviewer properly pointed out, our manuscript focuses on “time-varying” vaccination strategies. We have modified the text (e.g., avoiding any use of the word “reactive”) of the entire manuscript to clarify this concept.

Other comments:

7. The assumed contact matrix is likely to have a strong influence on the importance of different age-groups, when receiving an infection-blocking vaccine. It is good to see that this matrix is derived from local data; nonetheless, given the important underlying role of this matrix, there doesn’t appear to be enough consideration of how it drives the results. It would be good to see a sensitivity analysis to the assumed matrix coefficients (It would probably be enough simply to adopt contact matrices from other, contrasting settings). Similarly, the authors make assumptions for age-specific efficacy, that should also be subjected to additional analysis, given their obvious importance for the response of different age groups, to vaccination.

Response to comment #7: Once again, the Reviewer is correct. The contact matrix by age does have a strong impact in determining the prioritization strategies and the reduced incidences. Following the Reviewer’s suggestion, we have added a sensitivity analysis assuming different mixing patterns. In particular, we used a contact matrix derived for Shanghai where contact patterns refer to March 2020 (Zhang et al., Science Advances, 2021). At that time, the lockdown used to contain COVID-19 was over, but some interventions were still in place. This contact matrix highly differs from that used in the main analysis,

showing a much less assortative pattern by age. The contact matrix adopted in this sensitivity analysis is show Supplementary Fig. 23b. The results of this analysis are presented in the main text:

Line 99-103: *“However, if the vaccination program aims at minimizing the number of symptomatic cases or hospitalizations, the identified priority orders change when consider age-mixing patterns estimated during the pandemic (as compared to the pre-pandemic mixing patterns used in the baseline analysis; the obtained results are reported in Supplementary Fig. 19b).”*

Line 157-162: *“Although this increases the confidence in our findings, it is important to stress that the identified optimal strategies are sensible to variation in age-mixing contact patterns and differences in vaccine efficacy by age. This highlights the need to potentially adapt vaccination choices to the implemented NPIs (which may shape age-mixing patterns¹⁴⁻¹⁶) and the heterogeneity of vaccine efficacy across age groups.”*

Regarding vaccine efficacy by age. In the baseline analysis, we assumed an heterogenous age-specific efficacy based on Xia et al., Lancet Infect Dis, 2020. Following the Reviewer’s suggestion, we have now added a sensitivity analysis where we consider the vaccine efficacy to be homogenous by age. The results of this analysis are reported in Fig. 3c, Supplementary 19a, and commented in the main text as follows:

Line 103-106: *“Moreover, if the vaccine efficacy is identical across all age groups or if the vaccine is administered regardless of a previous history of infection, a high priority is given to adults aged over 65 years also when the vaccination campaign aims at minimizing the number of infections (e.g., Supplementary Fig. 19a and Supplementary Fig. 21a).”*

Line 157-162: *“Although this increases the confidence in our findings, it is important to stress that the identified optimal strategies are sensible to variation in age-mixing contact patterns and differences in vaccine efficacy by age. This highlights the need to potentially adapt vaccination choices to the implemented NPIs (which may shape age-mixing patterns¹⁴⁻¹⁶) and the heterogeneity of vaccine efficacy across age groups.”*

8. Please also give more information on the relative timing of the vaccination campaign, in relation to the epidemic. I believe at the moment, this timing is implicit in the initial conditions chosen for the epidemic (i.e. assuming that the vaccination programme is also initiated at time zero, a greater number of initial infecteds would mean a later vaccination programme in relation to the epidemic, and vice versa). For example, how would the results change if the vaccination is so early that it is completed before any substantial epidemic occurs? In this case, presumably there would be no benefit in priority sequencing. I realise this may be a lot of additional work to incorporate in the model itself, but at least some discussion of the point (if not outright modelling) would be helpful.

Response to comment #8: The reviewer’s understanding that vaccination campaign start at the initial of the epidemic is correct. We also would like to clarify that the optimal allocations (for the general population) start after essential workers (3.28% of the total population) are vaccinated. We fully agree with the Reviewer about the importance of the timing of vaccination campaigns relative to the epidemic onset. In the revised manuscript, we have added a set of scenario analyses to assess how the timing influences the prioritization strategies and effectiveness of the strategy.

First, we added a scenario where the vaccination program starts between 90 days before to 30 days after the epidemic onset. The obtained results are reported in Fig. 3g, Supplementary Figs. 7 and 13, and commented in the main text as follows:

Line 183-186: *“The timing of vaccination relative to the epidemic onset plays a crucial role in determining the effectiveness of prioritization strategies. Indeed, should a large enough fraction of*

the population already be vaccinated before an epidemic starts to unfold, the effectiveness of prioritized strategies is similar to that of a random vaccination (Fig. 3g and Supplementary Fig. 13)."

Moreover, we performed an analysis where the vaccine rollout is completed before the epidemic onset of epidemic. As pointed out by the Reviewer, in this case the prioritization becomes irrelevant (Supplementary Figure 22). We have added the following sentence to comment on it:

Line 106-108: *"Finally, should the vaccine rollout end before the onset of an epidemic, the prioritization order does not affect the final outcome, which entirely depends on the coverage (Supplementary Fig. 22)."*

REVIEWER COMMENTS

Reviewer #1 (Remarks to the Author):

I thank the authors for the careful attention given to the Reviewers comments and I am very appreciative of the fullness of the responses. A comprehensive amount of work has been undertaken that has immensely bolstered the robustness of the study findings.

Please find below a set of brief follow-up remarks.

In light of the comment raised by the other Reviewer clarifying the scope of 'dynamical' strategies, in which I am in agreement, the authors may like to consider amending the phrase "Dynamic optimization" in the manuscript title.

I would suggest a check of subpanel referencing in figure captions. For example, in Supplementary Fig. 21, it states "As panel c1 and d1 respectively," but there are no panels labelled c1 or d1.

For the line plots of the style shown in Figure 1, I would still recommend having distinct line types and/or markers for each line trace.

Reviewer #2 (Remarks to the Author):

The revised version has many new sensitivity analyses, that are welcome in substantially strengthening this study. Overall they show a quite intuitive picture - it's good to see, for example, how different mixing matrices might affect the results. I have only the following, remaining points of clarification.

I still have a concern about how mortality is modelled, even if the authors have added clarification that this is a post-hoc analysis. If I understand correctly, this basically means that, in the 'myopic' strategy described in the supporting information, it is assumed that anyone dying from infection essentially dies on the same day they got infected? (The myopic strategy must be making some assumption for mortality, since this is one of the various 'risk incidences' being used to make daily decisions.) This is clearly unrealistic, but intuitively, I don't think it would dramatically change the findings presented. It would just be good to see more discussion of this point in the main text. I would request the authors to mention this simplification upfront as one of the limitations; explain that it was done for tractability, since the myopic approach may not be able to cope otherwise; and if possible, also discuss reasons for why this assumption is probably not too important for the order of prioritisation.

I appreciate the clarifications the authors have made in the model equations, but another lingering concern is on the role of $v_i(t)$ in equations (1). Specifically, the fact that the recruitment of susceptible is proportional to $(S_i(t) - v_i(t))$. From the authors' explanation, I think I can see that this term aims to prevent anybody receiving vaccination at each time step, from being infected as a susceptible. However, there is already a term performing this function, which is the transition from $S_i(t)$ to $V_i(t)$. Are the equations in fact double-counting the reduction in susceptible by $v_i(t)$?

Line 131: The addition of the leaky vaccine scenario is a welcome one, but it's a surprise to see that the absolute, overall impact should be so low, in comparison with the all-or-nothing scenario. Are the authors able to shed some light on the reasons for this, perhaps in the discussion?

Line 164: I would recommend not treating 'supply' and 'rollout speed' as being equivalent. For example, even with an unlimited supply of vaccine, there will be a limit on the number of doses that can be given each day, because of limits on available staff to administer the vaccines, cold chain capacity at the vaccination points, etc. To avoid this confusion, I suggest replacing all references to 'supply' with 'vaccination rate' or equivalent.

Line 443, "with no increased efficacy in preventing the disease": I think the authors mean simply 'with no efficacy in preventing disease given infection', right?

Lines 446 to 447 suggest that people need two doses to get immunity - this seems to contradict elsewhere in the manuscript, where it's stated that you're focusing only on single-dose immunity. Please clarify.

Line 451, the denominator should read 'eigenvalues'

Several typos, e.g. line 214, "particularly relevance", need fixing

Responses to Reviewer's comments

Reviewer #1 (Remarks to the Author):

I thank the authors for the careful attention given to the Reviewers comments and I am very appreciative of the fullness of the responses. A comprehensive amount of work has been undertaken that has immensely bolstered the robustness of the study findings.

We would like to thank the reviewer for taking the time to assess our manuscript once again. We are glad to read that she/he believe the new analyses have “immensely bolstered the robustness” of our findings.

Please find below a set of brief follow-up remarks.

In light of the comment raised by the other Reviewer clarifying the scope of ‘dynamical’ strategies, in which I am in agreement, the authors may like to consider amending the phrase “Dynamic optimization” in the manuscript title.

As suggested, we have amended the title, which now reads “Time-varying optimization of COVID-19 vaccine prioritization in the context of limited vaccination capacity”.

I would suggest a check of subpanel referencing in figure captions. For example, in Supplementary Fig. 21, it states “As panel c1 and d1 respectively,” but there are no panels labelled c1 or d1.

Thank you for noticing this issue. We have revised the caption of Supplementary Fig. 19 (formerly Supplementary Fig. 21). We also went through each figure to double check all captions.

For the line plots of the style shown in Figure 1, I would still recommend having distinct line types and/or markers for each line trace.

Thank you for this suggestion which improves the readability of the figure. We have revised Figure 1 as suggested and apply the same change also to the other similar figures in the Supplementary Material.

Reviewer #2 (Remarks to the Author):

The revised version has many new sensitivity analyses, that are welcome in substantially strengthening this study. Overall they show a quite intuitive picture - it's good to see, for example, how different mixing matrices might affect the results. I have only the following, remaining points of clarification.

We would like to thank the reviewer for taking the time to assess our manuscript once again and the new set of comments. We are glad to read that she/he believe the new analyses have substantially strengthened our study.

I still have a concern about how mortality is modelled, even if the authors have added clarification that this is a post-hoc analysis. If I understand correctly, this basically means that, in the ‘myopic’ strategy described in the supporting information, it is assumed that anyone dying from infection essentially dies on the same day they got infected? (The myopic strategy must be making some assumption for mortality, since this is one of the various ‘risk incidences’ being used to make daily decisions.) This is clearly unrealistic, but intuitively, I don't think it

would dramatically change the findings presented. It would just be good to see more discussion of this point in the main text. I would request the authors to mention this simplification upfront as one of the limitations; explain that it was done for tractability, since the myopic approach may not be able to cope otherwise; and if possible, also discuss reasons for why this assumption is probably not too important for the order of prioritisation.

We apologize for the lack of clarity. When minimizing deaths, the myopic optimization accounts for the number of individuals that will eventually die among people that will die among those infected at the day t , regardless of when they actually die. In other words, in the optimization algorithm, we do not use the number of deaths at time t , but the expected number of deaths that will eventually occur among individuals that were infected at time t . We have added the following sentence to clarify our methodology:

“Line 489-491: For risk measures other than infections such as death, the myopic optimization accounts for the number individuals that will eventually die among those who were infected at the time t .”

Moreover, as properly pointed out by the reviewer, we do not expect that this will alter the order of prioritization and estimated effectiveness as shown in Supplementary Figure 14 (which was added in the previous round of revision) that accounts for uncertainties in the reporting rates and reporting delays of cases and was commented as follows: *“The identified prioritization strategies and the estimated effectiveness of the vaccination campaign are consistent when accounting for uncertainty in case reporting and delays (Fig. 3l, Supplementary Fig. 14).”*

I appreciate the clarifications the authors have made in the model equations, but another lingering concern is on the role of $v_i(t)$ in equations (1). Specifically, the fact that the recruitment of susceptible is proportional to $(S_i(t) - v_i(t))$. From the authors' explanation, I think I can see that this term aims to prevent anybody receiving vaccination at each time step, from being infected as a susceptible. However, there is already a term performing this function, which is the transition from $S_i(t)$ to $V_i(t)$. Are the equations in fact double-counting the reduction in susceptible by $v_i(t)$?

The Reviewer is correct that the term $(S_i(t) - v_i(t))$ is used to prevent individuals having administered with the first doses of the two-dose vaccines from get infected as susceptible from state S_i . However, we would like to clarify that i) At the beginning of each day t , v_i individuals in compartment S_i are instantly moved to compartment V_i where they can be infected; ii) The risk of infection for individuals in compartment S_i is the same as for those in compartment V_i as we assume that the first vaccine dose does not provide any protection. In light of these considerations, we believe we are not double counting the reduction in susceptible individuals. Nonetheless, to make sure of our interpretation, we have run the following experiment.

We have developed the classic epidemic model equivalent to the one proposed in our manuscript:

$$\begin{aligned}\dot{S}_i(t) &= -\beta \sum_j C_{i,j}^S \frac{I_j(t)}{N_j} S_i(t) - \bar{v}_i S_i(t) \\ \dot{V}_i(t) &= -\beta \sum_j C_{i,j}^S \frac{I_j(t)}{N_j} V_i(t) + \bar{v}_i S_i(t) - w V_i(t) \\ \dot{U}_i(t) &= -\beta \sum_j C_{i,j}^S \frac{I_j(t)}{N_j} U_i(t) + (1 - e_i) w V_i(t) \\ \dot{I}_i(t) &= -\beta \sum_j C_{i,j}^S \frac{I_j(t)}{N_j} [S_i(t) + V_i(t) + U_i(t)] - \gamma I_i(t) \\ \dot{R}_i(t) &= \gamma I_i(t) + e_i w V_i(t)\end{aligned}$$

where \bar{v}_i is the daily vaccination rate according to the classic interpretation given in mathematical epidemiology. All the other parameters and variable have the same meaning of those used in our baseline model used in the main text.

We have then run the random vaccination scenario using the baseline parameters given in the main text (i.e., $R=1.5$, 2 million first doses administered per day, all-or-nothing vaccine, pre-pandemic mixing patterns, etc.). We have then compared the outcome of this model with that obtained with the model used in our manuscript. The obtained results are very consistent between the two models (see figure below), with only small discrepancies given by the continuous nature of the classic model vs. the hybrid nature of our baseline model (which is continuous, except for the vaccination term that is instantaneous).

Line 131: The addition of the leaky vaccine scenario is a welcome one, but it's a surprise to see that the absolute, overall impact should be so low, in comparison with the all-or-nothing scenario. Are the authors able to shed some light on the reasons for this, perhaps in the discussion?

We thank the review for pointing it out. In light of her/his comment, we double check all the results for the leaky vaccine and realized that, although the code was correct, the ODE solver we used did not converged properly. After fixing the issue with the solver, the results of the leaky vaccine are close (although slightly worse) to those obtained with the all-or-nothing vaccine, exactly as expected by the reviewer. The new results are reported in Supplementary Fig. 12. For reviewer convenience, we report here Figure 2 (showing the results for the all-or-nothing vaccine) and Supplementary Fig. 12.

Figure 2. All-or-nothing vaccine

Supplementary Figure S12. Leaky vaccine

We have added the following sentence to the main text to comment these results:

“Lines 134-137: The “leaky vaccine” model identifies optimal prioritization strategies and effectiveness highly similar to those for the all-or-nothing model as well as quantitatively similar reductions of all risk incidences (Supplementary Figs. 7 and 12).”

We are truly indebted with the reviewer for this comment, and we highly appreciate her/his dedication to review our manuscript.

Line 164: I would recommend not treating ‘supply’ and ‘rollout speed’ as being equivalent. For example, even with an unlimited supply of vaccine, there will be a limit on the number of doses that can be given each day, because of limits on available staff to administer the vaccines, cold chain capacity at the vaccination points, etc. To avoid this confusion, I suggest replacing all references to ‘supply’ with ‘vaccination rate’ or equivalent.

We fully agree with the reviewer and apologize for the misleading wording. We have replaced the word ‘supply’ throughout the entire text.

Line 443, “with no increased efficacy in preventing the disease”: I think the authors mean simply “with no efficacy in preventing disease given infection”, right?

The reviewer is correct; we would like to thank her/him for noticing this improper wording. We have amended the text accordingly.

Lines 446 to 447 suggest that people need two doses to get immunity - this seems to contradict elsewhere in the manuscript, where it’s stated that you’re focusing only on single-dose immunity. Please clarify.

We apologize for the lack of clarity. We consider a 2-dose vaccine, where we assume that the first dose does not provide protection. We have now clarified at the beginning of the Results, where we mention for the first time the vaccination strategy, what we have simulated:

“Line 65-66: We consider 2.0 million first doses of a 2-dose vaccine administrated per day (0.14% rollout speed) [...]”

and

“Line 70-71: [...] and considering a 2-dose all-or-nothing vaccine where the first dose does not confer protection.”

Line 451, the denominator should read ‘eigenvalues’

Thank you; correction made.

Several typos, e.g. line 214, “particularly relevance”, need fixing

We carefully went through the text to fix the typos.

REVIEWER COMMENTS

Reviewer #2 (Remarks to the Author):

I convey my appreciation to the authors for substantially revising and improving the manuscript. They have addressed almost all of my previous points; I have only one remaining concern, relating to the equations (1) listed in the Methods.

As I commented earlier, the role of $v_i(t)$ in the term $(S_i(t) - v_i(t))$ is difficult to interpret. The authors have explained that this represents a number $v_i(t)$ being removed from the susceptible compartment at the beginning of each day, when they receive their first dose.

However, it seems to me that this transition is already modelled by the leading terms in the equations for dS/dt and dV/dt (i.e. $-v_i(t)$ and $v_i(t)$, respectively). Because individuals are being transferred from S to V through this term, then it seems there is no need to account for this transfer a second time, through the term $(S_i(t) - v_i(t))$.

It is helpful to see the comparison the authors have made between theirs and the 'classical' approach, but it is important that the equations should at least be mathematically correct. I suggest this could be achieved by simply replacing $(S_i(t) - v_i(t))$ by $S_i(t)$. [I believe this won't affect the model results, given that $v_i(t)$ seems to be operating independently of the ODE solver]

Responses to Reviewer's comments

Reviewer #2 (Remarks to the Author):

I convey my appreciation to the authors for substantially revising and improving the manuscript. They have addressed almost all of my previous points; I have only one remaining concern, relating to the equations (1) listed in the Methods.

We would like to thank the reviewer for taking the time to assess our manuscript once again and we are delighted to hear that she/he appreciated our work in “substantially revising and improving the manuscript”.

As I commented earlier, the role of $v_i(t)$ in the term $(S_i(t) - v_i(t))$ is difficult to interpret. The authors have explained that this represents a number $v_i(t)$ being removed from the susceptible compartment at the beginning of each day, when they receive their first dose.

However, it seems to me that this transition is already modelled by the leading terms in the equations for dS/dt and dV/dt (i.e. $-v_i(t)$ and $v_i(t)$, respectively). Because individuals are being transferred from S to V through this term, then it seems there is no need to account for this transfer a second time, through the term $(S_i(t) - v_i(t))$.

It is helpful to see the comparison the authors have made between theirs and the 'classical' approach, but it is important that the equations should at least be mathematically correct. I suggest this could be achieved by simply replacing $(S_i(t) - v_i(t))$ by $S_i(t)$. [I believe this won't affect the model results, given that $v_i(t)$ seems to operating independently of the ODE solver]

We thank reviewer for this comment. We have revised model equations as suggested as well as clarified the description of the objective function in the myopic optimal strategy. In light of the amended equations, we have rerun all the simulations in our analysis and updated the text and figures accordingly. As properly anticipated by the reviewer, our general conclusions remained unchanged.